# A new inventory of High Mountain Asia surging glaciers derived from multiple elevation datasets since the 1970s

Lei Guo[1], Jia Li[1], Amaury Dehecq[2], Zhiwei Li[1], Xin Li[3], Jianjun Zhu[1]

[1]School of Geo-science and Info-physics, Central South University, Changsha, 410083, China.
[2]Univ. Grenoble Alpes, IRD, CNRS, Grenoble INP, IGE, Grenoble, 38000, France.
[3]Institute of Tibetan Plateau Research, Chinese Academy of Sciences, Beijing, 100101, China.

*Correspondence to*: Jia Li (lijia20050710@csu.edu.cn)

**Abstract.** Glacier surging is an unusual undulation of ice flow and complete surging glacier inventories are important for regional mass balance studies and assessing glacier-related hazards. Glacier surge events in High Mountain Asia (HMA) are widely reported. However, the completeness of present inventories of HMA surging glaciers is constrained by the insufficient spatial and temporal coverage of glacier change observations, or by the limitations of the identification methods. In this paper, we established a new inventory of HMA surging glaciers based on the glacier surface elevation changes and morphological changes over four decades. Four kinds of elevation sources (KH-9 DEM, NASADEM, COP30 DEM, HMA8m DEM), three elevation change datasets, and long-term Landsat image series were utilized to access the distinctive change patterns of surging glaciers during two periods (1970s-2000 and 2000-2020). In total 890 surging and 336 surge-like glaciers were identified in HMA. Compared to the previous surging glacier inventories in HMA, our inventory incorporated more new surging glaciers. The number and area of surging glaciers accounted for ~2.49% (excluding glaciers less than 0.4 km$^2$) and ~16.59% of the total glacier number and glacier area in HMA, respectively. Glacier surges were found in 21 of the 22 subregions of HMA (except for the Dzhungarsky Alatau), however, the density of surging glacier is highly uneven. Surging glaciers are common in the northwest subregions (e.g., Pamir and Karakoram), but scarce in the peripheral subregions (e.g., Eastern Tien Shan, Eastern Himalaya, and Hengduan Shan). The inventory further confirmed that surge activity is more likely to occur for glaciers with larger area, longer length, and wider elevation range. Among the glaciers with similar area, the surging ones usually have steeper slope than the non-surging ones. Besides, we found a potential relationship between the surging glacier concentration and regional glacier mass balance. The subregions with slightly negative or positive mass balance hold large clusters of surging glaciers, while those with severe glacier mass loss hold very few surging glaciers. The inventory is available at: https://doi.org/10.5281/zenodo.7486614 (Guo et al., 2022).

**Key words:** High Mountain Asia, Surging glacier inventory, elevation change, KH-9, Digital Elevation Model (DEM)

## 1 Introduction

A surge is a glacier instability that translates into an abnormally fast flow over a period of a few months to years (Cogley et al., 2011). A surging glacier exhibits an active phase (surge) and a quiescent phase that may occur at quasi-periodic intervals (Jiskoot, 2011). While a glacier enters into the surging states, a large volume of ice mass is transported downstream at a higher-than-average speed. In the quiescence phase, a glacier stores to the slow-moving status again, and gradually regains mass at upper recaches. Previous studies pointed out that the surge-type glaciers only occupy ~1% of total glaciers (Jiskoot, 2011; Sevestre and Benn, 2015). However, glacier surges are far more than an occasional behavior in some specific regions, such as the Alaska-Yukon (Clarke et al., 1986), Svalbard (Jiskoot et al., 2000; Farnsworth et al., 2016), and Karakoram-Pamir (Bhambri et al., 2017; Goerlich et al., 2020; Guillet et al., 2022). Glaciers in these regions have experienced heterogeneous mass loss in the past decades (Hugonnet et al., 2021). How glacier surge activities impact the glacier regional mass balance needs further investigation, and to facilitate this kind of study, the glacier surges needed to be found out first.

In recent years, substantial efforts have been made to access the internal governing rules of glacier surges, including the
hydrological-control(Kamb, 1987; Fowler, 1987), thermal-control(Fowler et al., 2001; Murray et al., 2003), environmental
factor(Hewitt, 2007; Van Wyk de Vries et al., 2022), friction state(Thøgersen et al., 2019; Beaud et al., 2021), and the unified
enthalpy balance model (Sevestre and Benn, 2015; Benn et al., 2019). To support such studies, the accurate description of
surging glacier distribution is needed to provide samples for studying the internal dynamic process of surges. Besides, Glacier
surge can induce several kinds of hazards, e.g., glacier lake outbursts (GLOF) (Round et al., 2017; Steiner et al., 2018),
mudslides (Muhammad et al., 2021), or ice collapse (Kääb et al., 2018; Paul, 2019). Such mountain hazards have been
frequently reported in recent decades (Shugar et al., 2021; An et al., 2021; Kääb et al., 2021). A complete inventory of surging
glaciers is a basis for the regional hazard assessment of glacier surges.
Generally, a surging glacier could exhibit either one or several drastic changes, including: extreme speed-up (by a factor
10~1000 compared to normal conditions), distinct elevation change pattern, rapid terminus advance, and surface
morphological changes (medial or looped moraine, crevasses, etc.) (Jiskoot, 2011). The identification of surging glaciers can
be implemented based on the observation of the above changes, e.g., glacier surface morphology (Clarke et al., 1986; Paul,
2015; Farnsworth et al., 2016), terminus position (Copland et al., 2011; Vale et al., 2021), or glacier motion (Quincey et al.,
2011). As for the surge-type glacier, which refers to the glacier that possibly surged before, are generally identified by the
indirect morphological evidence (without observed changes) (Goerlich et al., 2020). The visual interpretation of glacier surface
morphological changes is easy to operate, but fraught with uncertainty due to the snow cover or the absence of supraglacial
moraine (Jacquemart and Cicoira, 2022). To recognize abnormal changes in glacier motion, a long-term flow velocity time
series is needed  (Yasuda and Furuya, 2015; Round et al., 2017). Since the quiescent phase may last for decades and the image
source for estimating the flow velocity is limited, the abnormal changes in glacier motion are prone to be missed. By contrast,
the recognition of abnormal surface elevation changes is an effective way to identify the surging glaciers, which has been
confirmed by several glacier mass-balance studies (Bolch et al., 2017; Zhou et al., 2018),  as its source datasets can satisfy the
requirement of spatial-temporal coverage with comparatively fewer acquisitions. By combining observations of multiple
features, the identification of surging glaciers could be more efficient and complete (Mukherjee et al., 2017; Goerlich et al.,
2020; Guillet et al., 2022). However, when conducting such studies on a large spatial scale or a long temporal scale, one should
select the least time-consuming but effective identification method. In that case, it's ideal to take the long-term elevation
change as the criteria,  and to combine with other observations as complements if possible (Guillet et al., 2022).
Except for the polar regions, High Mountain Asia (HMA) is the most densely glacierized region in the world. Within the HMA
range, several subregions are famous for the concentration of surging glaciers as well as the anomalous glacier mass balance
(Hewitt, 2005; Gardelle et al., 2013; Farinotti et al., 2020). The inventories of surging or surge-like glaciers have been
established for some subregions like the Karakoram (Bhambri et al., 2017), West-Kunlun (Yasuda and Furuya, 2015), Pamir
(Goerlich et al., 2020), Tien Shan (Mukherjee et al., 2017; Zhou et al., 2021). Sevestre and Benn (2015) presented the first
global surging glacier inventory by reanalyzing historical reports from 1861 to 2013. However, it was compiled from various
data sources (publications, reports, etc.) with inconsistent spatial-temporal coverage, which makes it difficult to ensure
accuracy and completeness. Vale et al. (2021) identified 137 surging glaciers across HMA by detecting surge-induced terminus
change and morphological changes from Landsat images from 1987 to 2019. The number is obviously underestimated, because
it is smaller than the numbers of previous subregional inventories (Bhambri et al., 2017; Goerlich et al., 2020). Guillet et al.
(2022) presented a new surging glacier inventory of HMA by identifying multiple glacier change features. In total 666 surging
glaciers were identified across HMA. However, the glacier change observation period is shorter than two decades (2000-2018),
and therefore some surging glaciers with relatively long revisit cycles may be missed.
In this study, we aimed to build a new inventory to include more surging glacier within HMA based on glacier surface elevation
change observations over four decades. A workflow was developed to obtain the historical glacier surface elevation change
from multiple datasets, including the KH-9 DEM (1970s), NASADEM (2000), COP30 DSM (2011-2014), HMA8m DEM
(2002-late 2016), and existed elevation change datasets. Glaciers in the new inventory were divided into three classes of
confidence in surge detection. After that, the elevation change based inventory were further complete and corrected by the
long-term timeseries morphological feature identification based on Landsat images (1986-2021). Based on the present
inventory, the distribution and geometric characteristics of surging glaciers within HMA were statistically analyzed, in order
to demonstrate their spatial heterogeneity and geometrical difference from the normal glaciers.

## 2 Study region

High Mountain Asia consists of the Qinghai-Tibet Plateau and the surrounding regions, including the Karakoram, Pamir,
Himalayas, and Tien Shan. According to the updated Glacier Area Mapping for Discharge from the Asian Mountains
(GAMDAM2) glacier inventory, HMA hosts 131819 glaciers, covering a total area of ~99817 km$^2$ (Sakai, 2019). The Hindu
Kush Himalayan Monitoring and Assessment Programme (HiMAP) divided HMA into 22 subregions (Fig. 4) (Bolch et al.,
2019). Different subregions are influenced by different air currents, such as the South Asia monsoon, the East Asia monsoons,
and the westerlies (Bolch et al., 2012; Maussion et al., 2014). Glacier mass balance across HMA was found to be heterogeneous
in the past decades (Gardelle et al., 2013; Brun et al., 2017; Shean et al., 2020). In particular, glaciers in the Pamir-Karakoram-
West Kunlun region had a slightly positive or balanced mass budget (Hewitt, 2005; Zhou et al., 2017; Farinotti et al., 2020),
while those in the Eastern Himalayas, Nyainqentanglha and Hengduan Shan mountain ranges experienced substantial ice loss
(Maurer et al., 2019).

## 3 Datasets

### 3.1 Elevation Data

The NASADEM is mainly reprocessed from the C-band SRTM (Shuttle Radar Topography Mission) images. Among the
current global DEMs, the NASADEM has the shortest source data acquisition period (~11/02/2000~22/02/2000) (Farr et al.,
2007). Based on an improved production flow, the NASADEM has a better performance than the earlier SRTM void-free
product in most regions (Crippen et al., 2016). The NASADEM was employed as the reference elevation source because its
acquisition time, 2000, is suitable to divide the elevation change observations to before and after 21st century with moderate
time span (one or two decades). Each tile of the product has an extent of 1°× 1° and a pixel spacing of 1 arc-second (see Fig.
1a). In total 313 tiles were downloaded from NASA LP DAAC
(https://e4ftl01.cr.usgs.gov/MEASURES/NASADEM_HGT.001/).
Another global DEM we utilized is the newly released Copernicus DEM GLO-30-DGED (i.e., COP30 DEM). The COP30
DEM was edited from the delicate WorldDEM™, which was generated based on the TanDEM-X mission. The global RMSE
of COP30 DEM is ±1.68 m (AIRBUS, 2020). Several studies have pointed out that this DEM is the most reliable open-access
DEM to date (Purinton and Bookhagen, 2021; Guth and Geoffroy, 2021). The source images of COP30 DEM were mostly
acquired between 2011 and 2014, and therefore COP30 DEM is suitable to represent the surface elevation in the 2010s. Like
the NASADEM, the COP30 DEM has a pixel spacing of 1 arc second. Each tile of product has an extent of 1°× 1°. In total
313 tiles were downloaded through ESA Panda (https://panda.copernicus.eu/web/cds-catalogue/panda).
The High Mountain Asia 8-meter DEM (HMA8m DEM) was also utilized in this study. The HMA8m DEM was generated
from high-resolution commercial optical satellite stereo images, including WorldView-1/2/3, GeoEye-1, and Quickbird-2
(Shean et al., 2020), through an automated photogrammetry workflow that is integrated with multiple error-control processes
(Shean et al., 2016). This DEM was originally produced for the mass balance estimation of HMA glaciers, so it covered most
of the glacierized regions in HMA. In total 3598 DEM tiles were downloaded from National Snow and Ice Data Center
(https://nsidc.org/data/HMA_DEM8m_MOS/versions/1). About 95% of them were acquired between 2010 and 2016 (Fig. 1b).
Due to the data voids and inconsistent acquisition time, the HMA8m DEM was taken as a supplementary elevation source to
increase the observations in the 2010s.
The Hexagon KeyHole-9 (KH-9) imagery was acquired in the 1970s. It is one of the earliest near-global satellite stereo image
source. The KH-9 imagery is characterized by a spatial resolution of 6-9 m, a wide coverage (130 km x 260 km), and a 70%
forward overlap (Surazakov and Aizen, 2010). Many studies have utilized this imagery to estimate historical glacier surface
elevation (Holzer et al., 2015; Zhou et al., 2017; Maurer et al., 2019). The KH-9 DEMs used in this study were generated
through the automated ASPy pipeline (Dehecq et al., 2020). The methodology, validated in the European Alps and Alaska
achieved a vertical accuracy of ~5m (68% confidence level). For more details on the method of KH-9 DEM generation, please
refer to Dehecq et al. (2020). In total 238 DEMs with a resolution of 48 m were generated from the KH-9 images acquired
between 1973 and 1980. The KH-9 DEMs were utilized to represent the glacier surface elevation in the 1970s (See Fig. 1c).
Several newly published elevation change datasets were also collected to include the most recent surges as much as possible
(Brun et al., 2017; Shean et al., 2020; Hugonnet et al., 2021). We mainly used the elevation change results presented by
Hugonnet et al. (2021) to extend the observation period to 2020, which has a resolution of 100 m and a temporal interval of 5
years. Through the inter-comparison of the multiple elevation change results, the gross errors or false signals in the elevation
change patterns could be easily detected.

### 3.2 Optical Satellite Images

In order to assist the identification of surging glacier, we also recognized the glacier morphological feature changes from multi-
temporal optical satellite images. The 1986-2021 Landsat imageries were mainly utilized to capture the glacier morphological
changes. We downloaded the false-colour composited Landsatlook images (geo-referenced) that have good brightness contrast
over snow/ice areas from USGS website (https://earthexplorer.usgs.gov). The images were pre-selected to satisfy the
requirement of cloud cover (<10%). In total, 7843 Landsatlook images in 148 frames were used (see Fig. 1d). We also utilized
the very high-resolution (VHR) images (Google/ESRI/Bing, etc.) as complements for surging feature identification. The fine
resolution of these images allows us to visually check the possible morphological features caused by past surges.

### 3.3 Glacier inventory

In this study, we used the GAMDAM2 glacier inventory (Sakai, 2019) as template for the surging glacier inventory, rather
than the Randolph Glacier Inventory V6.0 (RGI6.0) (RGI Consortium, 2017). The GAMDAM glacier inventory has included
many small glaciers that are missed in RGI6.0, and provides a more accurate glacier extent by excluding outcrop rocks and
shaded areas (Nuimura et al., 2015). Since the GAMDAM2 inventory only contains the glacier polygon vectors, we calculated
the geometric and topographic attributes for each glacier in a way similar to that of RGI6.0. The maximum glacier centreline
was calculated through the Open Global Glacier Model (OGGM) (Maussion et al., 2019). The attributes were used to interpret
the geometric characteristics of surging glaciers.

### 4 Methodology

### 4.1 Estimation of glacier surface elevation change

The four kinds of DEMs have different coordinate references, vertical references, and data formats. Firstly, all DEMs were
converted to float GeoTiff format. For datasets with quality files (NASADEM and the COP30 DEM), the DEM were
preprocessed to mask out the pixels of low quality. The poor pixels of COP30 DEM tile were determined through the attached
height error map (with values larger than 2.5 m) and water body map (with values not equal to zero). The NASADEM was
directly masked with the attached water mask file. Subsequently, the coordinate system, map projection, and vertical reference
of all DEMs tiles were unified as the WGS84 coordinate system, HMA Albers Equal Area projection (Shean et al., 2020), and

WGS84 ellipsoid. The glacier surface elevation changes during 2000-2010s were derived by subtracting the NASADEM from the COP30 DEM and HMA8m DEM, and those during 1970s-2000 were derived by subtracting the KH-9 DEM from the NASADEM.

An automated DEM differencing workflow for large-scale glacier surface elevation change estimation was developed based on the *demcoreg* package presented by Shean et al. (2019). The workflow integrated multiple DEM co-registration approaches, the polynomial fit of tilt error, and other adaptive outlier removal approaches that was operated based on the observations over stable regions. Hence, a mask that excluded the water bodies and glacierized regions was generated in advance. Before differencing, the two DEMs need to be co-registered, because a small geolocation shift can result in considerable elevation change errors in high mountain regions. The efficient analytical DEM co-registration method presented by Nuth and Kääb (2011) was used to eliminate the relative geolocation shift between DEMs. This method assumes the geolocation shift vectors of all DEM pixels are identical. However, for the global DEM products like NASADEM and COP30 DEM, a DEM tile was usually mosaiced from multiple DEM patches, and the geolocation shift vectors at different parts of the DEM tile may be different. In view of this problem, we developed a block-wise version of the analytical DEM co-registration method to reduce the impacts of geolocation accuracy anisotropy of a DEM tile. Each DEM tile was divided into m×n blocks, and the DEM shifts were estimated for each block. Then, the m×n groups of shift parameters were merged into one group of shift parameters through a cubic interpolation. Technically, the estimated shift parameters become increasingly representative as the block size decreases. However, the fitting of shift parameters requires a certain number of samples. The final block size was set to 300×300 pixels to reach the best balance between the representativeness and estimation accuracy of shift parameters. Besides, we found that the block-wise co-registration method could result in wrong fitting of shift parameters over flat regions. To deal with this, a threshold of mean slope (10°) was set to classify the DEMs into the flat and the hilly categories, and the original global co-registration method (Nuth and Kääb, 2011) was applied to the flat ones.

Due to the residual orbital error of satellite images, the elevation difference maps often showed planimetric trends. This type of systematic error was fitted as a universal surface trend using a quadratic polynomial model based on the observations in stable regions, and then was removed from the elevation difference tile (Li et al., 2017). Besides, due to the jitter of the SAR antenna and optical mapping camera, the elevation difference maps often showed stripes (i.e., band-like artifacts) (Yamazaki et al., 2017). To eliminate the stripes, the elevation difference map was converted to the frequency domain through the Fast-Fourier-Transform method. Since the cyclic values have a high frequency in the power spectral density map, a threshold of frequency was set to separate the stripes components from the normal elevation differences. The de-stripping was completed after the backward transformation. Finally, the outliers of elevation difference maps were reduced through the 3-sigma threshold criteria.

The radar penetration into glacier surface can result in biases of elevation change estimation, which could be several to dozens of meters, and potentially can lead to the false positive identification. We adopted a two-step procedure to reduce the radar penetration bias in the final elevation change results. First, we used the DEM differencing workflow mentioned above to minus the NASADEM from the SRTM-X DEM. The elevation differences over glacierized area were regarded as the penetration difference between X-bands and C-bands. Secondly, we fitted a 3rd polynomial function between the glacial dH and altitude, which was deemed as the penetration depth – altitude relationship. Then, the radar penetration biases were removed from the COP30 DEM related results by taking the glacier elevation as input for the function. For the dH results calculated by differencing NASADEM and optical DEMs (e.g. HMA8m and KH-9 DEM), the penetration difference of X- and C- bands was multiplied by 2 to represent the absolute penetration depth of C-band (Abdel Jaber et al., 2019; Fan et al., 2022) and then removed from the related results.

Finally, three elevation change maps were calculated: the COP30 DEM – NASADEM, the HMA8m DEM – NASADEM, and the NASADEM – KH-9 DEM. The first two elevation change maps were combined with the three elevation change datasets for surging glacier identification during the period 2000-2020, and the last one during the period 1970s-2000. In total, our

elevation change observations covered ~92% of the total glacier area within HMA in 2000-2020, and ~77% in 1970s-2000.
Gaps in observations were mainly due to: 1) data voids and incomplete coverage of original DEMs tile, which was the main
cause for the KH-9 DEMs and HMA8m DEM related results; 2) gross error removal during the elevation change calculations,
which led to the scattered holes in the COP30 DEM related results.

**4.3 Surging glacier identification**

The identification of surging glaciers in this study were divided into three steps. First, we generated a raw inventory of surging
glaciers through the qualitative interpretation of multi-temporal elevation changes. Then, the visual identification of
morphological feature changes was carried out for the identified surging and surge-like glaciers. This procedure can further
confirm the surges or correct the false identifications based on glacier elevation changes (Guillet et al., 2022). The identified
results were re-checked by careful inspection on VHR images, and by comparing with existed surging glacier inventory. Also,
the surging tributaries were separated from the non-surging glacier trunk at this step.

**4.3.1 Identification through elevation changes**

In general, a typical glacier surge cycle can be divided into three phases (Jiskoot, 2011): 1) the build-up phase, characterized
by remarkable thickening in the upper reaches; 2) the active phase, characterized by remarkable thinning in the upper reaches
and thickening in the lower reaches; 3) the post-surge phase, characterized by strong down-wasting in the lower reaches. The
classical method of identifying surging glaciers is to recognize the combination of marked upper thinning and lower thickening
in the longitudinal direction. However, to distinguish the surging glaciers in the build-up or post-surge phase, careful
comparison with surrounding glaciers is required, which is difficult to be carried out with a mathematical index. In this study,
we established a three-class indicator to distinguish the surge possibility through the visual interpretation of glacier elevation
change patterns:
I) "verified":

- a) obvious thickening in lower reaches (e.g. +30 m);

- b) contrasting upper-thinning (e.g. +20 m) and lower-thickening (e.g. +20 m);

- c) contrasting upper-thickening (e.g. +20 m) and lower-thinning (e.g. -30 m);

- d) severe thinning in the lower reaches (two time stronger than that of the normal glaciers, or comparable to the

ablation of adjacent "verified" surging glaciers);

II) "probable":

- a) moderate upper thinning (e.g. -15m) and lower thickening (e.g. +15m);

- b) only moderate thickening in the middle reaches (e.g. +15m);

III) "possible":

- a) only moderate thickening at the terminus (e.g. +15m);

- b) only strong thinning in the lower reaches (one time stronger than adjacent normal glaciers).

Note that, the specific values of elevation change mentioned above were for information only. Because of the diversity in the
regional elevation change patterns under different climate or topographic conditions, the thresholds may vary spatially.
The identification of surging glaciers was conducted separately in the two observation periods (1970s-2000 and 2000-2020).
The sub-inventory covering the period 1970s-2000 was generated based on the dH results of NASADEM – KH-9 DEM. For
the sub-inventory covering the period 2000-2020, its dH datasets contain the COP30 DEM – NASADEM, the HMA8m DEM
– NASADEM, and three previously published elevation change datasets (Brun et al., 2017; Shean et al., 2020; Hugonnet et
al., 2021). Within each observation period, each glacier will be labelled with its possibility level of surging and elevation
change pattern in the attribute table. For example, the label of "I-c" means this glacier was classified as "verified" surging
glaciers because contrasting upper-thickening and lower-thinning pattern were observed in the corresponding period. Figure 2
shows an example of surging glacier identification result.

### 4.3.2 Identification through morphological feature changes

Long-term Landsat images (acquired between 1986 and 2021) were utilized to investigate the morphological change features
of the three types of potential surging glaciers identified from elevation change. With each Landsat image acquisition frame,
all Landsatlook images of different dates (acquired from 1986 to 2021) were be merged into an animated time-series image.
Based on the animated image, we are able to easily identify the morphological feature changes. Regarding the moderate
resolution of Landsat images, only three types of feature changes were utilized as criteria for identifying glacier surges:
terminus position change, looped moraine changes, and medial moraine changes. Similarly, we assigned a two-level index to
each morphological change to indicate our confidence at the identification, which was defined as follow:
1) terminus advance:

I) : obvious terminus advancing (e.g.  over 500 m);

II): small terminus advancing ( e.g.  0~500 m);

2) looped/medial moraine change:

I) : fast formation/vanishment of the looped moraine, or obvious distortion of the medial moraine;

II) : slow formation or vanishment of the looped moraine, or slight shape changes of existed looped moraine, or

slight distortion of the medial moraine.

Each of the three kinds of morphological changes were individually qualified and labelled in the attribute table.

### 4.3.3 Generation of surging glacier inventory

Through the above identification steps, in total five indicators were compiled to describe the changes of possible surging
glaciers. The two sub-inventories of dH identified results were merge firstly followed the principle of possibility, i.e.,  if a
glacier was identified as a surging glacier in both two periods but attached with different indicators, its indicator in the final
inventory was taken from the indicator having a higher possibility. The possibility of indicators follows the order: "verified" >
"probable" > "possible". For example, a glacier was identified as a "verified" surging glacier in the period 1970s-2000, and
was identified as a "probable" surging glacier in the period 2000-2010s, then it was qualified as a "verified" surging glacier.
After that, the merged dH indicators were further compared with the morphological indicators to determine the final indicator
of surge possibility. The "probable" or "possible" class was changed to a class with higher possibility(e.g., from "probable" to
"verified" ) only if a "I" kind of morphological change was found.
We think the advancing glaciers usually have such features: 1) only thickened in a small area at terminus, without contrasting
upper thinning; 2) the advancing distance is relatively short (Lv et al., 2019, 2020; Goerlich et al., 2020). These features are
corresponding to the "III-a" type of elevation change, and "II" type of terminus advance. Therefore, if a glacier only shows
these two kinds of changes, it will be qualified as an advancing glacier, rather than a surging glacier.
For some glacier complexes that only tributary surged while the trunk did not, such as the Biafo glacier, Fedchenko Glacier
and Panmah Glacier (Hewitt, 2007; Goerlich et al., 2020; Bhambri et al., 2022), it's necessary to separate the surging tributary
from the trunk. A tributary will be considered as an individual surging glacier if it has the following features. Firstly, the
dividing line of contrasting elevation change locates within this tributary. Secondly, the volume of mass contributed by this
tributary to the glacier trunk is relatively small. Then we manually edited the outline to separate the tributary from the glacier
complex. This kind of surges was also marked by the attribute of "trib_surge".
In the final step, we inspected the identified surging glaciers on VHR imagery. The inspection aimed to remove the wrong
identification due to some false signals, such as the severe lower-thinning in a lake-terminated glacier and remarkable surface
heightening caused by nearby landslide. We also refined our inventory through the careful comparison with inventories
presented by Guillet et al. (2022), Goerlich et al. (2020) and Bhambri et al. (2017). For the surging glaciers that identified in
other inventories but not included in ours, we did a careful re-identification.

## 5 Results

### 5.1 Identified surging glaciers

A total of 1226 surge-related glaciers across the HMA were identified based on the elevation changes and morphological
feature changes. The identified surge-related glaciers consisted of 890 'verified' surging ones, 208 'probable' ones, and 128
'possible' ones. A total of 175 surging tributaries were identified in 86 glacier complexes. When merging the identification
results of the two periods, we found that a considerable proportion of identified surging glaciers were simultaneously
recognized in two periods. This makes our inventory more convincible, since a surging glacier could exhibit different kinds of
changes in different periods. For example, 26 probable and 51 possible surging glaciers identified during 2000-2020 turned to
be "verified" surging glaciers during 1970s-2000. Meanwhile, 60 "probable" and 21 "possible" surging glaciers identified
during 1970s-2000 turned out to be 'verified' surging glaciers during 2000-2020. Thanks to an almost complete coverage of
the elevation change observations, we were able to classify all glaciers in HMA. Table 1 shows the number of surging glaciers
identified from two periods of elevation changes and morphological feature changes. Due to the incomplete coverage and data
voids of KH-9 DEMs, fewer surging glaciers were identified during the period 1970s-2000. The "probable" and "possible"
classes were deemed as surge-like glaciers. To avoid confusion, only the "verified" surging glaciers were used for analysis
and comparison.

### 5.2 Distribution of surging glaciers

Surging glaciers were identified in 21 subregions of HMA (except for the Dzhungarsky Alatau), however, the density of
identified surging glaciers is far from even (Fig. 3). Glacier surges are common in the northwest regions, sporadic in the inner
regions, and scarce in the peripheral regions. Figure 4 and Table 2 show the ratios of surging glacier number and area in each
subregion. Considering the area of the smallest identified surging glacier is 0.42 km$^2$, we only took the glaciers larger than
0.40 km$^2$ in the glacier number related ratio. The surge-like glaciers were not accounted in such statistics. The number (890)
and area (16556.42 km$^2$) of identified surging glaciers accounted for ~2.49% and ~16.59% of the total glacier number and
glacier area in HMA, respectively.
Among the 22 subregions, the Karakoram is the largest cluster of surging glaciers. In total 354 surging and 128 surge-like
glaciers were identified in the Karakoram. The number and area of verified surging glaciers in the Karakoram accounted for
39.80% and 47.90% of the total glacier number and area within HMA. In the Karakoram, surging glaciers has accounted for
8.59% of the total glacier number. We found more than half of the tributary surges (101) in the Karakoram, where large glaciers
are much more developed than other regions. The area of surging glaciers occupied 39.48% of the total glacier area in
Karakoram. The Pamirs, composed of the Eastern Pamir, Western Pamir and Pamir Alay, hosts 249 surging glaciers and 128
surge-like glaciers. About 27.74% of the glacier area in the Eastern and Western Pamir belongs to surging glaciers. We also
found 28 surging tributaries in 15 glacier complexes in the Pamirs. Surging glaciers are also common in the Western Kunlun.
In total 82 surging and 47 surge-like glaciers were identified in the West Kunlun, and the area of surging glaciers accounted
for 30.48% of the total glacier area. The Central Tien Shan has the fourth largest surging glacier area. In total 59 surging
glaciers were identified in the Central Tien Shan, which covered 12.93% of the total glacier area. The Karakoram, Pamirs,
West Kunlun, and Central Tien Shan nourished ~83% of the surging glaciers across HMA. Figure 5 shows the distribution of
identified surging and surge-like glaciers in these four regions.
Within interior HMA subregions (including the Tibetan Interior Mountains, Eastern Kunlun Shan, and Tanggula Shan), the
number of identified surging glaciers only covered less than 2% of the total glacier number, but the area accounted for near
15% of the total glacial area. Surging glaciers in these regions generally gathered in some watersheds. Similar localized surging

glacier clusters were also found in the Nyainqentanglha, Northern and Western Tien Shan, and Central Himalaya, but the corresponding area ratios are much lower. In these regions, our inventory covered dozens of surging glaciers which were rarely reported before. Figure 6 shows some samples of identified surging glaciers in these regions.

**5.3 Geometric characteristics of surging glaciers**

In this part, only the surging glaciers and non-surging- glaciers are taken for analysis. The surge-like glaciers are not included. All glacier samples in the surging and non-surging classes are larger than 0.40 $km^2$.

We divided all glaciers into 9 classes according to their area, and calculated the ratios of surging glacier number and area in each class. As shown in Figure 7, surging glaciers were found in all classes. Both the ratios of surging glacier area and number became increasingly high as the glacier size increased, except for the last class. Surging glaciers with an area of 1~50 $km^2$ occupies 82% of all surging glaciers. For the three classes in which glaciers are larger than 50 $km^2$, the ratios of surging glaciers area and number were about 52% and 54%, respectively. In particular, 2 of 6 very large glaciers (the Siachen glacier and Hispar glacier) surged during our observation periods.

When comparing the geometric characteristics of the surging glaciers and non-surging glaciers, we selected samples in the following way: for each surging glacier, we selected 10 non-surging glacier samples that have closest area; and then we randomly sampled 3 out of the 10 selected non-surging glaciers. This is to minimize the discrepancy resulted from the sample differences. There are two reasons for doing so. First, the gap between the sample numbers is huge (~35000 non-surging vs. 890 surging). Second, a high proportion of non-surging glaciers are very small glaciers. The final selected 890×3 non-surging glaciers formed the reference group.

We first analysed the distribution of surging glacier number and area in eight orientations. As shown in Fig. 8, both the number and area of glaciers facing the north are the largest, and then followed by those facing the northwest and northeast. The distribution of the glacier orientation in reference group were different than that of the non-surging glaciers, which confirmed the statistical analysis would be affected by sample differences. The number of surging glaciers facing the north accounted for ~30.1% of the total surging glacier number, and their area accounted for ~27.8% of all surging glacier area. The number and area ratios of surging glaciers facing the north are obviously higher than that of the non-surging glaciers facing the north, while the number and area ratios of surging glaciers facing the northwest are obviously lower than that of the non-surging glaciers facing the northwest. Meanwhile, the area ratio of surging glaciers facing the northeast is considerable higher than the number ratio, but for surging glaciers facing the northwest and southwest the situation is opposite.

Figure 9 illustrates the comparisons between the basic geometric properties of surging and non-surging glaciers. The sampling strategy mention above was also utilized here. If we directly compare the surging glaciers with all non-surging glaciers, we will find that surging glaciers generally have a larger area, wider elevation range (i.e., the highest glacier surface elevation minus the lowest), and longer flowline (Fig 9a-c). Taking the median values as the candidates s, the quantitative comparisons are 7.3 $km^2$ (surging) vs. 0.87 $km^2$ (non-surging) for glacier area, 1534 m vs. 642 m for elevation range, and 6695 m vs. 1854 m for maximum glacier length, respectively. In terms of mean surface slope and median elevation, the values of the surging glaciers are less spread out than the non-surging glaciers. However, the median values of the two kinds of glaciers are very close (see Figures 9d and 9e). If we took the non-surging glaciers in reference group for comparisons, the discrepancies of two kinds of groups on these geometric properties became much more different. The gaps between the surging and non-surging glaciers (reference group) in the glacier area (7.3 $km^2$ vs. 7.0 $km^2$), elevation range (1534 m vs. 1180 m) and glacier length (6695 m vs. 5560 m), are much smaller. More importantly, the mean slope of the glaciers in reference group become smaller than that of the surging glaciers.

The correlation between different glacier geoqetric properties was analysed through the bivariate scatterplots (see Figure 10). Among the glacier area, glacier length, and glacier surface elevation range, any two of them have an apparent positive correlation. The glacier mean slope has a moderate correlation with the glacier area, glacier length, and glacier elevation range.

By contrast, the glacier median elevation has little correlation with glacier area, glacier length, glacier elevation range, and glacier mean slope. The correlation of any two geometric properties makes little difference between surging and non-surging glaciers.

## 6 Discussion

### 6.1 Uncertainty analysis

The reliability of surging glacier identification is directly related to the accuracy of glacier surface elevation change. Assuming the uncertainties in surface elevation change are similar over glacierized areas and stable areas, we evaluated the glacier elevation change uncertainties based on elevation change observations in stable areas, whose true values are zeros. Meanwhile, the uncertainties in the radar penetration calculation were also considered through the error propagation law. The normalized median absolute deviation (NMAD) is less sensitive to outliers and can be deemed as an alternative to standard deviation (Höhle and Höhle, 2009). Hence, the NMAD was used to denote the uncertainty of individual glacier surface elevation change tile (Li et al., 2017). Figure 11 shows the NMAD of elevation change observations in stable areas within each DEM differencing tile, which were used for the co-registration and biases removal during the glacier elevation change estimation. Due to large distortions in the KH-9 images, the NASADEM - KH-9 DEM results had the highest uncertainties. Benefiting from the advantages of bistatic SAR image pairs, the COP30 DEM has high quality, and the COP30 DEM related results had the lowest uncertainties. The HMA8m DEM related results had moderate uncertainties. The average NMAD of all DEM differencing tiles was smaller than 5 m. The significant elevation errors usually occurred in the highly rugged regions such as crests and horns. The terrain of glacier surface is relatively gentle, and therefore the uncertainties of glacier surface elevation changes should be lower than the estimated values. In general, the uncertainties of our elevation change results are well-controlled. Compared with the typical surface elevation change resulted from a glacier surge (tens to hundreds of meters), the magnitudes of uncertainties are very small.

Similar to previous studies (Sevestre and Benn, 2015; Goerlich et al., 2020), the surging glacier identification in this study was completed through a manual qualitative interpretation way. It's difficult to provide a quantitative index to represent the uncertainty of surge identification. However, the four-class indicator of surge likelihood could aid that in a degree.

### 6.2 Characteristics of surging glaciers

The direct comparisons between geometric characteristics of surging and non-surging glaciers manifest that surge activity is more likely to occur in the glacier with a larger area, wider elevation range, and longer length (Fig. 9). Previous studies also reported this phenomenon (Barrand and Murray, 2006; Jiskoot, 2011; Sevestre and Benn, 2015; Mukherjee et al., 2017; Guillet et al., 2022). Larger area, wider elevation range, and longer length mean a larger glacier scale and more mass storage. Surge is a self-balancing process of a glacier to regulate its internal instability of thermal or hydrologic conditions which needs enough mass storage. In this case, about 97% of the surging glacier has an area of larger than 1 $km^2$. For glaciers larger than 10 $km^2$, surge becomes a quite common behavior (with a number ratio higher than 20%), rather than an accidental behavior (see Fig.7).

In terms of mean surface slope, we could not observe a statistically difference in the median value of the surging and non-surging glaciers, although the surging glaciers have a more concentrated value range (Fig 9d and Figure 10, 3rd row, 1st column). After minimizing this kind of bias, we observed an obvious higher mean slope of surging glaciers in the comparison with the reference group. Several studies have demonstrated that the surging glacier tend to have shallower slope (Jiskoot et al., 2000; Guillet et al., 2022). However, here we reasonably argue that this rule was concluded from an unbalanced comparison, as the non-surging glaciers are consist of much larger proportion of small glaciers than surging glaciers does. Meanwhile, the inverse relationship between the glacier slope and length (Clarke, 1991; Sevestre and Benn, 2015) may not apply to very small

glaciers (i.e. smaller than 1 km$^2$). As shown in Fig. 9d and Fig. 10, among the non-surging glaciers, the small ones occupy a high proportion and their mean slope presents strong variability. Regarding this, we can conclude that steeper glaciers are more likely to surge when the comparison is restricted to similar areas. Considering the fact that steeper valley glaciers are more prone to reach the crucial gravitational imbalance, this conclusion should be reliable. As for the glacier median elevation, since it is almost irrelevant to the glacier area, glacier length, glacier elevation range, and glacier mean slope (see Fig. 10), it can be deemed as an irregular glacier index. However, among glaciers that have similar areas, steeper glaciers generally have lower median elevation. That's why the median elevation of surging glaciers is slightly smaller than that of non-surging glaciers (Fig. 9e).

These comparisons could now lead to a conclusion as follows: the surging glaciers are generally longer, and have larger elevation spanning than non-leapfrog glaciers, since they have more mass storage. However, when glaciers are similar in area, a steeper surface slope is more likely to lead to surge.

Besides, our results manifested that the ratio distribution of surging glaciers in eight aspects are slightly different from that of non-surging glaciers (see Fig. 8). This is in line with the findings in previous studies (Bhambri et al., 2017; Goerlich et al., 2020). In particular, the ratio of surging glaciers is relatively higher than the non-surging glaciers in the north direction, but lower in the northwest direction. This is mainly caused by the orientation of the mountains in Karakoram and Pamir. It is generally known that glaciers facing the north are more developed in HMA. Due to the orientation of the mountains, most of the large glaciers in Karakoram and Pamir flow toward the north and northeast. The number of large glaciers flowing towards the northwest is much less. Accordingly, the surging glaciers facing the north and northeast are much more than that facing the northwest (see Fig. 5). The number of surging glaciers in Karakoram and Pamir accounts for a considerable proportion of the total number of surging glaciers in HMA, and therefore the orientation of surging glaciers there has a great impact on the orientation distribution of surging glaciers in HMA.

The spatial distribution of surging glaciers in HMA presents strong heterogeneity. About 83% of identified surging glaciers were located in the northwest region including the Central Tien Shan, Pamirs, Karakoram, and West Kunlun, and their area occupied about 87% of the total identified surging glacier area (see Fig. 4 and Table 2). As discussed above, larger glaciers are more likely to surge. The northwest regions generally hold more large glaciers, and therefore hold more surging glaciers. In other subregions, large glaciers are usually concentrated in some great ice fields, such as the Geladandong, Puruogangri, and Xinqingfeng. Accordingly, surging glaciers in these subregions are usually clustered in several watersheds.

Several studies have pointed out that glacier surge activities have little impact on the glacier mass balance (Gardelle et al., 2013; Bolch et al., 2017; Guillet et al., 2022). However, glacier mass balance may also affect the occurrence of glacier surge. Copland et al. (2011) concluded that the increase of glacier surges in the Karakoram could be related to the positive mass budget. The accumulated ice mass would accelerate a glacier to surge (Eisen et al., 2005; Kochtitzky et al., 2020), and the significant mass loss could prevent or postpone the surge in return (Dowdeswell et al., 1995). On a regional large scale, the relationship between mass balance and surge occurrence needs to be further analysed. Our glacier elevation change maps of the period 2000-2010s are similar to that derived by Brun et al. (2017) and Shean et al. (2020). We found that, at the regional scale, the occurrence of surging glaciers is correlated with the regional glacier mass balance. The three subregions holding the largest clusters of surging glaciers, i.e., the Pamirs, Karakoram, and West Kunlun, are characterized by slightly negative or positive mass budgets, which is known as the 'Pamir-Karakoram-West Kunlun' anomaly (Brun et al., 2017). Likewise, the subregions Central Tien Shan, Tibetan Interior Mountains, and East Kunlun Shan, which hold the moderate clusters of surging glaciers, have glacier mass loss rates much lower than the average rates of HMA. By contrast, subregions with severe glacier mass loss hold the lowest surging glacier ratio, such as the Dzhungarsky Alatau, Hengduan Shan, and Eastern Himalaya.

## 6.3 Comparison with previous surging glacier inventories

Guillet et al. (2022) presented a comprehensive surging glacier inventory of HMA for the period 2000-2018 from a multi-factor remote sensing approach. Prior to the comparison, we generated an inventory based on the RGI6.0, as Guillet et al. (2022) did. Guillet et al. (2022) identified 666 surging glaciers, and the area of surging glacier occupies 19.5% of the total glacier area. We identified 890 surging glaciers (809 if RGI6.0 was used), and their area only occupies 16.59% of the total glacier area. We attributed the lower area ratio of surging glaciers to two reasons. First, in our inventory the surging tributaries were separated from the non-surging trunks. Second, many outcrop rocks and shaded areas are excluded from the GAMDAM2 glacier areas (Sakai, 2019), which would lower our surging area ratio, but make the result more accurate. If we assign our identified surging glaciers to the RGI6.0 polygons without tributary separation, the surging area ratio would be larger (20.25%). Within our inventory, 556 surging and 62 surge-like glaciers were also identified by Guillet et al. (2022), and the discrepancy of identifications mostly occurred on small glaciers. If only the period 2000-2020 was considered, 657 surging glaciers were identified by us, which is very close to that of Guillet et al. (665). For the period 1970s-2000, there are 151 surging and 101 surge-like glaciers that were not identified by Guillet et al. (2022). Overall, we have newly identified 253 surging and 248 surge-like glaciers. We owed the newly findings to the longer observation period and multiple elevation change observation. However, 47 surging glacier presented by Guillet et al. were missed in this study, and 62 surge-like glaciers in our new inventory were identified as surging glaciers by Guillet et al. (2022). We carefully checked the glaciers not included in our inventory but included in Guillet et al's inventory, as well as those included in our inventory but not included in Guillet et al's inventory, and this step helped us to find 21 more surging glaciers. We attribute this to the deficiency of using a single criterion, which could be aided by combining other features. Besides, the DEMs used in this study were suffering from the data voids and incomplete spatial coverage, especially for the KH-9 DEM, which could result in a relatively conservative identification. Multiple studies have identified surging glaciers in the Karakoram based on different data sources. For example, Bhambri et al. (2017) identified 221 surging and surge-like glaciers (the tributaries of a glacier system are counted as individual glaciers) based on the glacier morphology changes detected from spaceborne optical images acquired from 1972 to 2016, in-situ observations, and archive photos since the 1840s. However, the boundary used by Bhambri et al. (2017) to define the extent of Karakoram is much smaller than that used in our inventory. A much smaller group of surging glaciers (88) were identified by Copland et al. (2011) based on a similar method and the data acquired between 1960 and 2013. Rankl et al. (2014) identified 101 surging glaciers in the Karakoram by detecting the changes in glacier surface velocity and terminus position between 1976 and 2012. The results of Guillet et al. (2022) should be more reliable than previous ones, because more criteria were used for identifying surging glaciers. Compared with previous inventories, our inventory includes more surging glaciers (354). Among the 223 surging glaciers in the Karakoram identified by Guillet et al. (2022), 203 were identified as surging glaciers, and 12 were identified as surge-like glaciers in this study, which means only 8 surging glaciers presented by Guillet et al. (2022) were not included in our inventory. The high coincidence between the two inventories indicates our surging glacier identification result is reliable. In total, we have newly identified 101 surging and 101 surge-like glaciers in this region.

In the Pamirs, Sevestre and Been (2015) identified 820 surge-type glaciers based on publications and reports, but Goerlich et al. (2020) reported only 186 surging glaciers based on the observations of glacier flow velocity, elevation change, etc.. We found that, if Goerlich et al. (2020) applied the GAMDAM2 glacier polygons used in this study, the number of identified surging glaciers should be 182. Among the 182 surging glaciers identified by Goerlich et al. (2020), 153 and 15 were identified as surging and surge-like glaciers in our study, respectively. Although 14 surging glaciers are missed in this study, our inventory has contained other 94 surging and 44 surge-like glaciers. The main cause for the result discrepancy is that the glacier elevation change observation conducted by Goerlich et al. (2020) only covered parts of the Western Pamir and only the observations before 2000 were used. In this region our inventory shared 193 surging glaciers with Guillet et al's inventory, and 185 of them were identified during the period 2000-2020, which also manifests a high coincidence of the two results.

In the West Kunlun, Yasuda and Furuya (2015) reported 9 surging glaciers in the main range only, based on changes in glacier
flow velocity and terminus position of 31 glaciers, and other 9 surging glaciers were found in the northwest part of the West
Kunlun Shan by Chudley et al. (2019). A larger number (60) were found by Guillet et al. (2022). However, our inventory has
even included more surging (82) and surge-like (47) glaciers in the West Kunlun Shan. During the period 2000-2020, we have
identified 61 surging glaciers, which is very close to the number presented by Guillet et al. (2022). In Central Tien Shan,
Mukherjee et al. (2017) identified 39 surge-type (9 surging and 30 surge-like) glaciers through the analysis of changes in
surface elevation and morphology from 1964 to 2014, whereas 79 (59 surging and 20 surge-like) were identified in our studies.
The insufficient coverage of elevation change observation (only covered the west part of the Central Tien Shan) may be the
main reason for the discrepancy in identification results. Guillet et al. (2022) identified 54 surging glaciers during 2000-2018,
in which 36 were confirmed in our inventory.
**7 Conclusions**
This study presented a new inventory of surging glaciers across the entire HMA range, which was accomplished based on the
glacier surface elevation changes derived from multiple elevation sources, by using the morphological feature changes from
optical images as complements. In total 890 surging and 336 surge-like glaciers were identified in the new inventory. Through
the analysis of geometric parameters, we found that surging glaciers generally have a greater area, length, and elevation range
than non-surging glaciers. However, the differences are smaller than we thought if taking the glacier size distribution into
account. When glaciers having similar area, the steeper one is more likely to surge. Furthermore, combing the region-wide
glacier mass balance measurements, we found a similar distribution between the positive mass balance and number of surging
glaciers. Benefiting from the long period and wide coverage of surface elevation change observations, our study identified
much more surging glaciers in HMA than in previous studies. However, our inventory does not provide the surge duration
period and the maximum flow velocity to describe the dynamic process of each glacier surge activity. Improvements should
be made by combining multi-criteria identification methods. Considering the fact that glacier surges are more widespread than
we thought, the inventory presented in this study still needs further replenishment.
**8 Data and code availability**
The presented inventory is freely available at: https://doi.org/10.5281/zenodo.7486614 (Guo et al., 2022). The dataset is
composed of two files including the inventory itself and the associated metadate file. The inventory is distributed in the format
of GeoPackage (.gpkg) and ESRI shpfile (.shp), which is represented by outline or centroid of surging glaciers with geometric
attributes. The glacier polygons of the inventory are compiled from the GAMDAM2 glacier inventory. In total eight fields are
integrated in the attributes table to describe the surging information of corresponding glacier as mentioned in section 4.3. The
description of each field in the attribute table is listed in Table 3. The metadata file is stored in a text file (README.txt),
which contains the description and details of the  attributes information of the inventory.
The code used for elevation change estimation can be available at: https://github.com/TristanBlus/dem_coreg. This code was
developed based on the *demcoreg* package (Shean et al., 2019).
**Author contribution**
J.L. and L.G. conceived this study and wrote the paper. L.G. developed the processing flow, complied the inventory and drew
the figures with the support from J.L. A.D. generated the KH-9 DEM. A.D., Z.L. and X.L. helped with the results analysis and
discussions and manuscript editing. Z.L., J.L. and J.Z. provided the funding acquisition. All authors have contributed and
agreed to the published version of the manuscript.
**Competing interest**
The authors declare that they have no conflict of interest.
**Acknowledgments**
The authors express gratitude to all institution that provide us the opensource dataset used in this study: the NASADEM from
LP DAAC (https://e4ftl01.cr.usgs.gov/MEASURES/NASADEM_HGT.001/), the Copernicus DEM from Eruopean Space
Agency (ESA) (https://spacedata.copernicus.eu/web/cscda/cop-dem-faq)., the HMA8m DEM processed by David Shean from
National Snow and Ice Data Center (NSIDC) (https://nsidc.org/data/HMA_DEM8m_MOS/versions/1), and the Randolph
Glacier Inventory Version 6.0 (http://www.glims.org/RGI/randolph.html). The authors also appreciate the valuable comments
from Frank Pual and Guillet Gregoire.
**Financial support**
This work was supported by the Strategic Priority Research Program of Chinese Academy of Sciences (XDA20100101), the
National Natural Science Foundation of China (41904006), the National Natural Science Fund for Distinguished Young
Scholars (41925016), the Hunan Key Laboratory of remote sensing of ecological environment in Dongting Lake Area (No.
2021-010), the Fundamental Research Funds for the Central Universities of Central South University (2021zzts0265).

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

## Tables and Figures

**Tabel 1: Surging glacier identification results**

| Glacier changes | Identification class | | | Total |
|---|---|---|---|---|
| | I | II | III | |
| 2000-2020 elevation change | 719 | 157 | 169 | 1045 |
| 1970s-2000 elevation change | 507 | 156 | 57 | 720 |
| 1986-2021 terminus advance | 247 | 397 | - | 645 |
| 1986-2021 looped moraine | 112 | 31 | - | 144 |
| 1986-2021 medial moraine | 69 | 29 | - | 108 |
| **Final identified surging glaciers** | 890 (verified) | 208 (probable) | 128 (possible) | 1226 |

**Tabel 2: Results of surging glacier identification in 22 subregions of HMA. Only glaciers larger than 0.4 km² were considered in the glacier number related values.**

| HiMAP regions | Glacier Number | | | | Glacier Area | | | |
|---|---|---|---|---|---|---|---|---|
| | Surging | Surge-like | Total | Ratio (%) | Surging | Surge-like | Total | Ratio (%) |
| Karakoram | 354 | 128 | 4121 | 8.59 | 7936.12 | 1329.40 | 20103.68 | 39.48 |
| Western Pamir | 188 | 48 | 3058 | 6.15 | 2232.52 | 289.597 | 8172.64 | 27.32 |
| Western Kunlun Shan | 82 | 47 | 2508 | 3.27 | 2580.21 | 589.17 | 8466.12 | 30.48 |
| Central Tien Shan | 59 | 20 | 2248 | 2.62 | 881.61 | 305.47 | 6816.95 | 12.93 |
| Eastern Pamir | 56 | 16 | 1148 | 4.88 | 796.35 | 79.12 | 2746.47 | 29.00 |
| Tanggula Shan | 22 | 4 | 697 | 3.16 | 441.94 | 41.71 | 1937.39 | 22.81 |
| Tibetan Interior Mountains | 22 | 12 | 1471 | 1.50 | 286.29 | 140.22 | 3933.48 | 7.28 |
| Northern Western Tien Shan | 21 | 6 | 1374 | 1.53 | 116.27 | 81.09 | 2502.60 | 4.65 |
| Central Himalaya | 17 | 21 | 3433 | 0.50 | 164.12 | 185.07 | 9928.72 | 1.65 |
| Eastern Kunlun Shan | 16 | 7 | 1191 | 1.34 | 458.11 | 55.38 | 2960.26 | 15.48 |
| Nyainqentanglha | 10 | 5 | 2916 | 0.34 | 119.53 | 184.79 | 7216.62 | 1.66 |
| Eastern Hindu Kush | 9 | 5 | 1279 | 0.70 | 178.18 | 77.19 | 3055.80 | 5.83 |
| Western Himalaya | 9 | 4 | 3659 | 0.25 | 110.22 | 69.41 | 8619.19 | 1.28 |
| Eastern Himalaya | 6 | 0 | 1334 | 0.45 | 94 | 0 | 3371.89 | 2.79 |
| Pamir Alay | 5 | 0 | 991 | 0.50 | 35.72 | 0 | 1957.94 | 1.82 |
| Qilian Shan | 4 | 6 | 851 | 0.47 | 35.99 | 26.40 | 1627.94 | 2.21 |
| Eastern Tibetan Mountains | 3 | 2 | 156 | 1.92 | 36.33 | 3.85 | 341.46 | 10.64 |
| Altun Shan | 2 | 3 | 156 | 1.28 | 4.13 | 3.17 | 294.95 | 1.40 |
| Eastern Tien Shan | 2 | 1 | 1243 | 0.16 | 12.03 | 2.59 | 2440.11 | 0.49 |
| Hengduan Shan | 2 | 0 | 700 | 0.29 | 26.22 | 0 | 1335.39 | 1.96 |
| Gangdise Mountains | 1 | 0 | 768 | 0.13 | 10.52 | 0 | 1339.54 | 0.79 |
| Dzhungarsky Alatau | 0 | 1 | 407 | 0 | 0 | 10.98 | 648.61 | 0 |
| Total | 890 | 336 | 35709 | 2.49 | 16556.42 | 3474.60 | 99817.72 | 16.59 |

* The value of ratio only considered the number and area of surging glaciers.

**Table 3: Attribute information in the present surging glacier inventory.**

| Attribute | Description | Attribute | Description |
|---|---|---|---|
| | | | |

| Glac_ID | Glacier identifier composed by Lat/Lon | Surge_20 | Surge identified in 2000-2020 by dH |
|---------|----------------------------------------|----------|--------------------------------------|
| Area | Glacier covered area (km$^2$) | Surge_70s | Surge identified in 2000-2020 by dH |
| Zmin | Minimum elevation of the glacier (m a.s.l) | Delta_T | Identified class of glacier terminus advance |
| Zmax | Maximum elevation of the glacier (m a.s.l) | Loop_M | Identified class of looped moraine change |
| Zmed | Median elevation of the glacier (m a.s.l) | Medial_M | Identified class of medial moraine change |
| Slope | Mean glacier mean surface slope (°) | False_signal | False positive signal of identification |
| Aspect | Mean glacier aspect/orientation (°) | Trib_surge | If the glacier has/is surging tributary |
| MaxL | Maximum length of glacier flow line (m) | Surge_class | Final surge identification during 1970s-2020 |
| HiMAP_region | HMA subregion that the glacier belongs to | | |


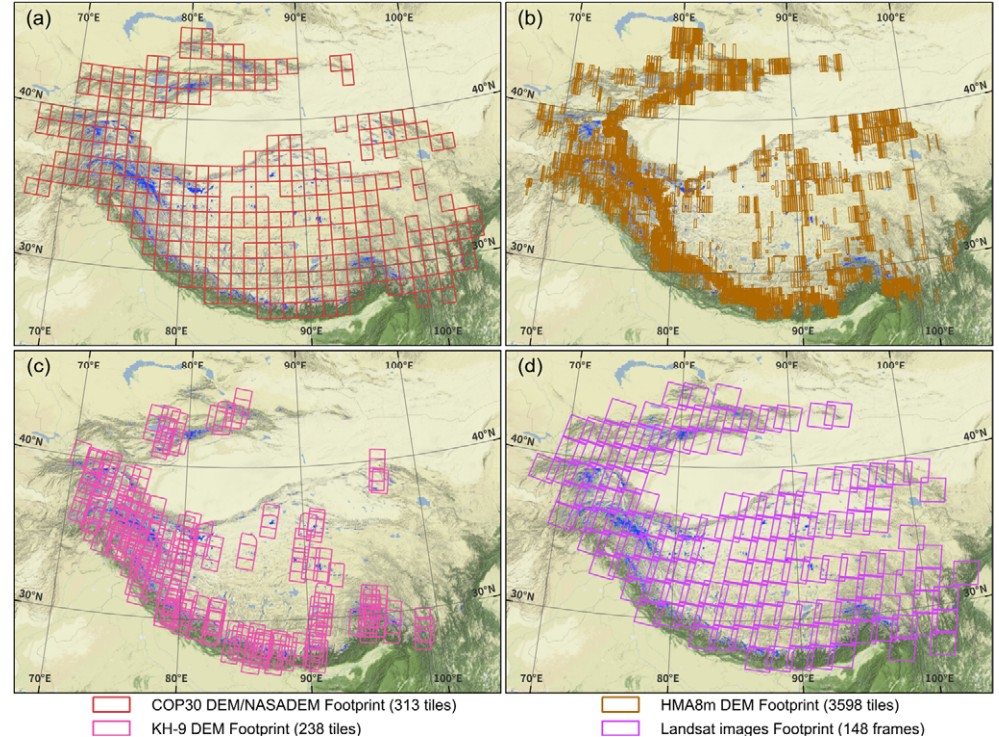

COP30 DEM/NASADEM Footprint (313 tiles)    HMA8m DEM Footprint (3598 tiles)
KH-9 DEM Footprint (238 tiles)    Landsat images Footprint (148 frames)


**Figure 1: Footprints of (a) COP30/NASA DEMs, (b) HMA8m DEMs, (c) KH-9 DEMs and (d) Landsat imageries that were utilized in this study. The background is rendered from the ESRI World Physical base map (Source: US National Park Service).**

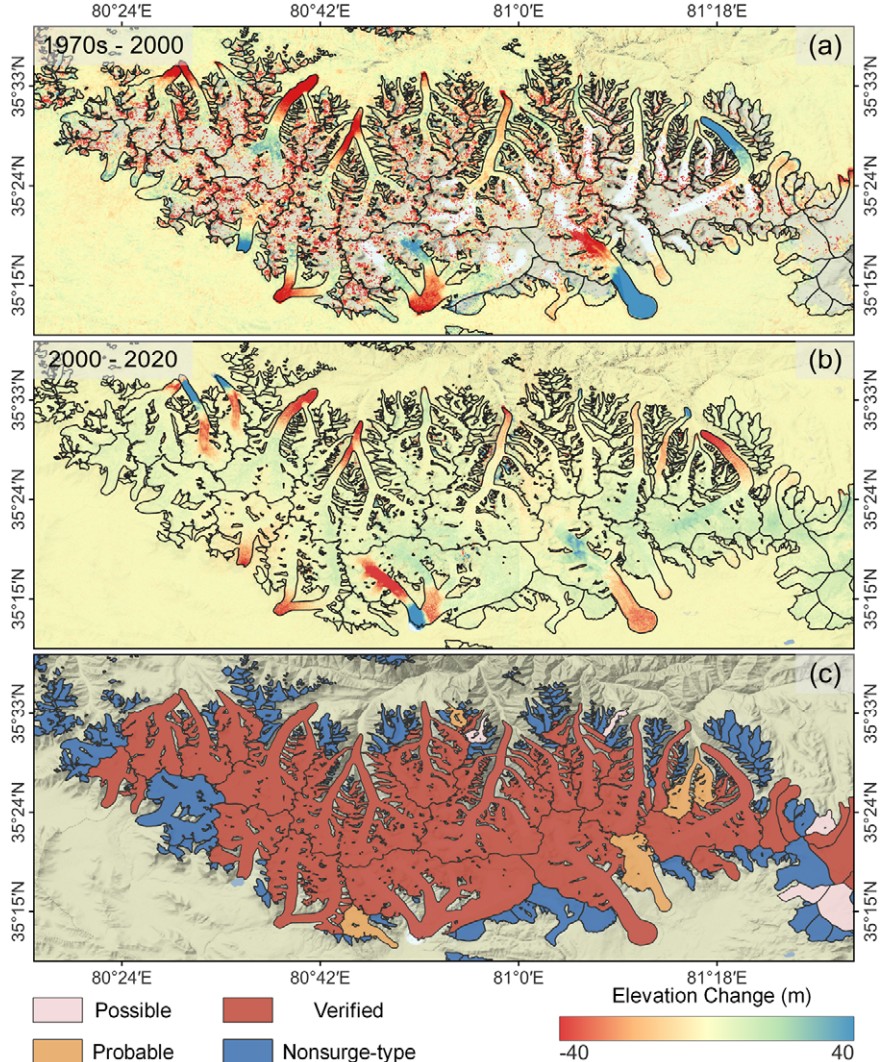

**Figure 2: An example of derived elevation change maps during 1970s-2000 (a) and 2000-2020 (b), and the corresponding surging glacier identification result (c). Black curves are glacier outlines. The background is the shaded relief of COP30 DEM (Source: ESA). The area is in the main massif of Western Kunlun Shan.**

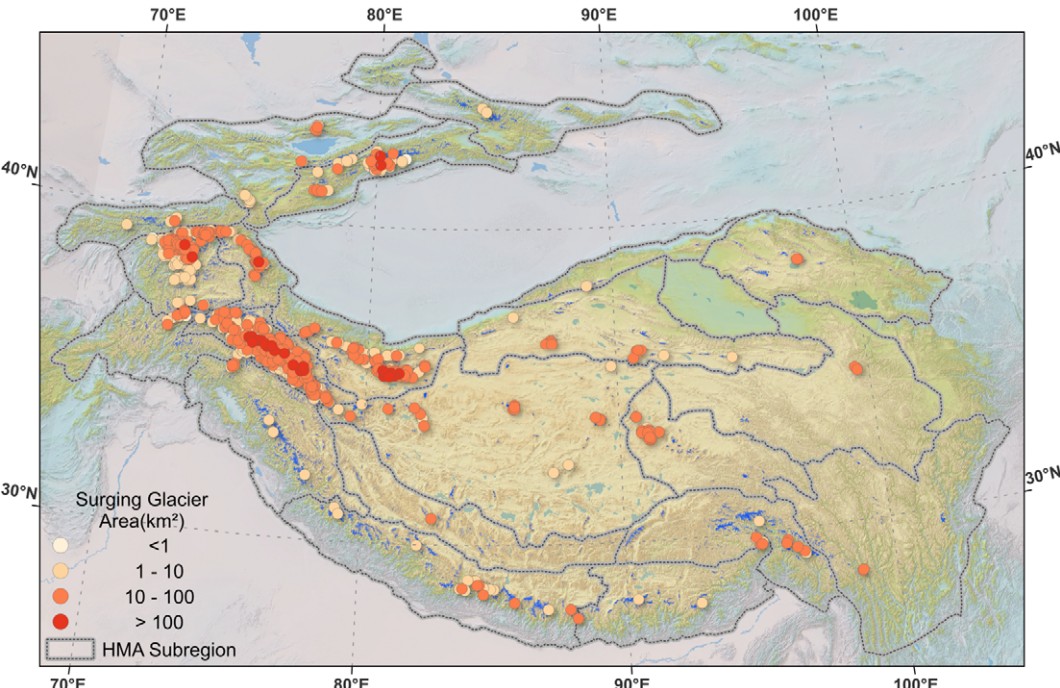

**Figure 3: Overview of the distribution of identified surging glaciers in 22 subregions of HMA. The background is the shaded relief of SRTM DEM (Source: USGS).**

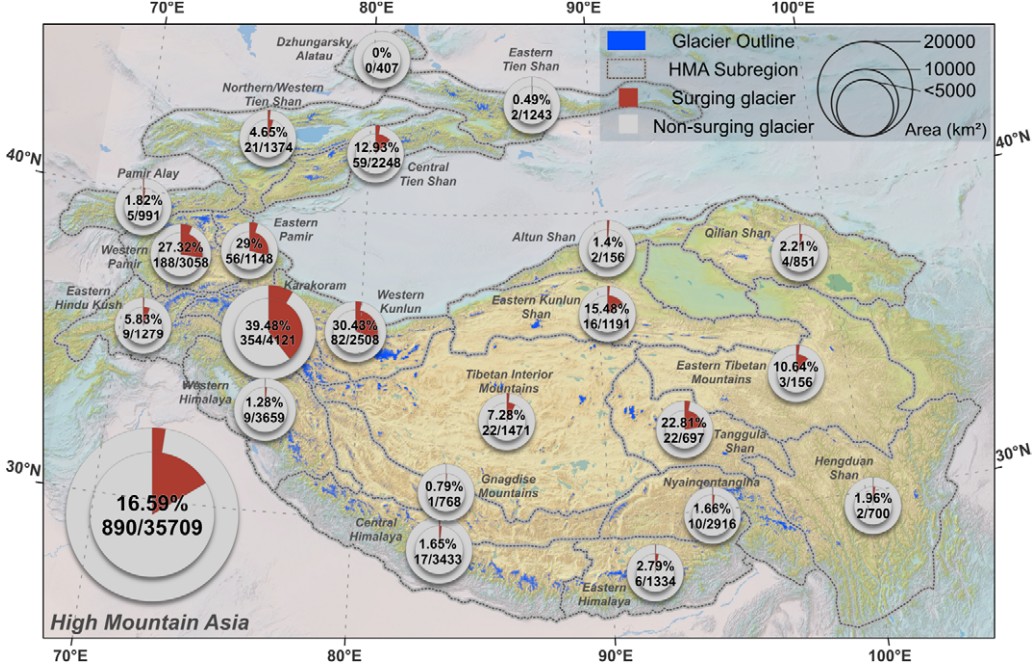

Figure 4: Distribution of surging glaciers in the 22 subregions of HMA. The double-level pie chart represents the ratios of surging glacier number and area in each subregion. The inner pie denotes the area ratio labelled by a percentage, and the outer pie denotes the number ratio labelled by a fraction (only considered glacier larger than 0.4 km²). The background is the shaded relief of SRTM DEM (Source: USGS).

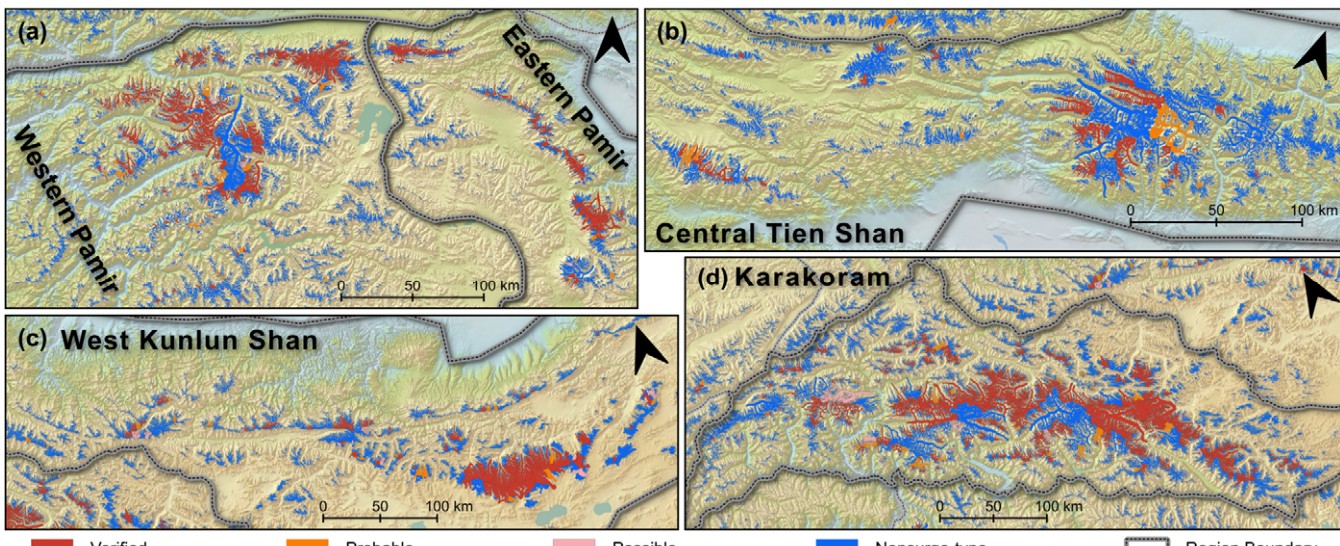

Figure 5: Results of surging glacier identification in the Pamirs (a), Central Tien Shan (b), West Kunlun Shan (c), and Karakoram (d). The background is the shaded relief of SRTM DEM (Source: USGS).

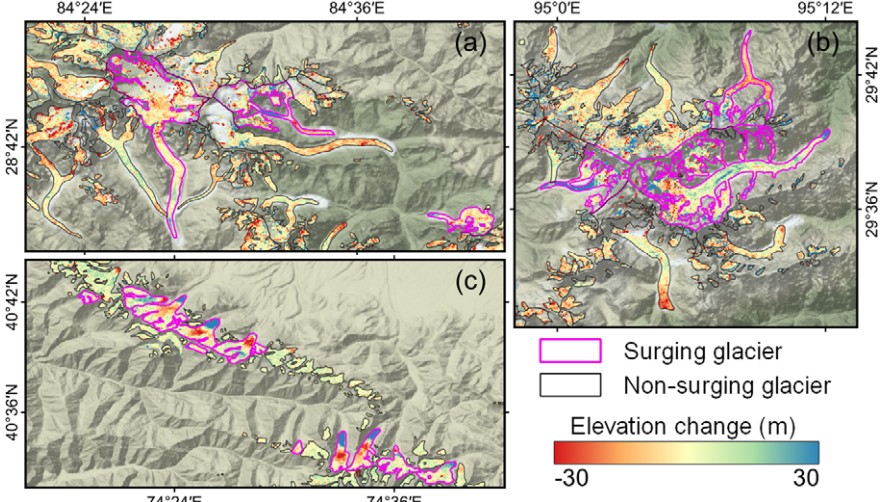

Figure 6: Elevation change map of identified surging glaciers samples in (a) Central Himalaya (1970s-2000) , (b) Nyainqentanglha (1970s-2000), and (c) Northern Western Tien Shan (2000-2020). Background is the shaded relief of SRTM DEM (Source: USGS).

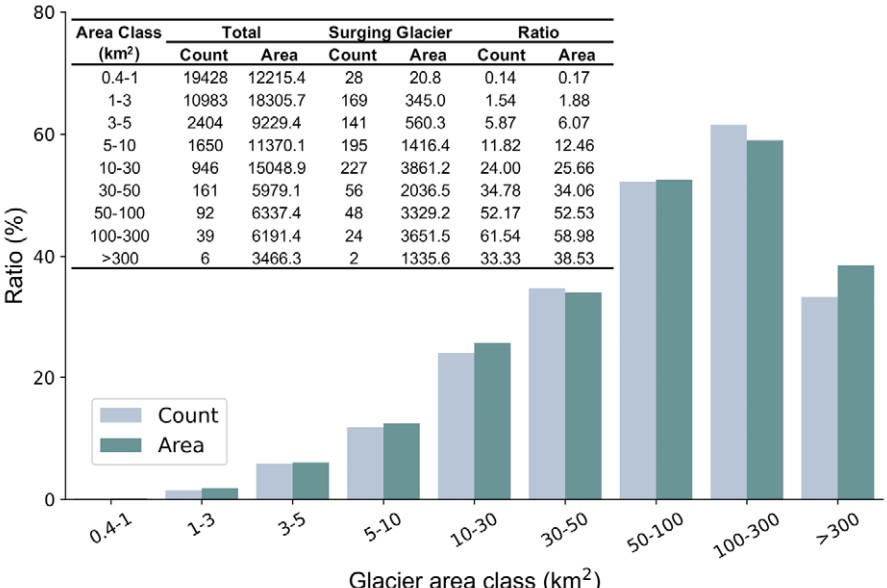

| Area Class (km²) | Total | | Surging Glacier | | Ratio | |
|---|---|---|---|---|---|---|
| | Count | Area | Count | Area | Count | Area |
| 0.4-1 | 19428 | 12215.4 | 28 | 20.8 | 0.14 | 0.17 |
| 1-3 | 10983 | 18305.7 | 169 | 345.0 | 1.54 | 1.88 |
| 3-5 | 2404 | 9229.4 | 141 | 560.3 | 5.87 | 6.07 |
| 5-10 | 1650 | 11370.1 | 195 | 1416.4 | 11.82 | 12.46 |
| 10-30 | 946 | 15048.9 | 227 | 3861.2 | 24.00 | 25.66 |
| 30-50 | 161 | 5979.1 | 56 | 2036.5 | 34.78 | 34.06 |
| 50-100 | 92 | 6337.4 | 48 | 3329.2 | 52.17 | 52.53 |
| 100-300 | 39 | 6191.4 | 24 | 3651.5 | 61.54 | 58.98 |
| >300 | 6 | 3466.3 | 2 | 1335.6 | 33.33 | 38.53 |

Figure 7: The ratios of surging glacier number and area in different classes.

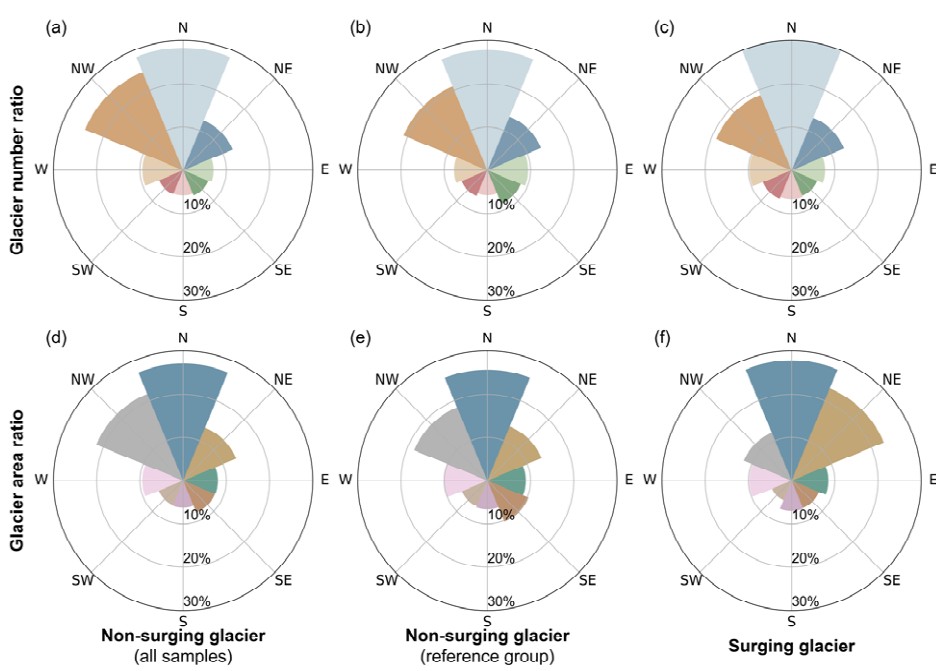

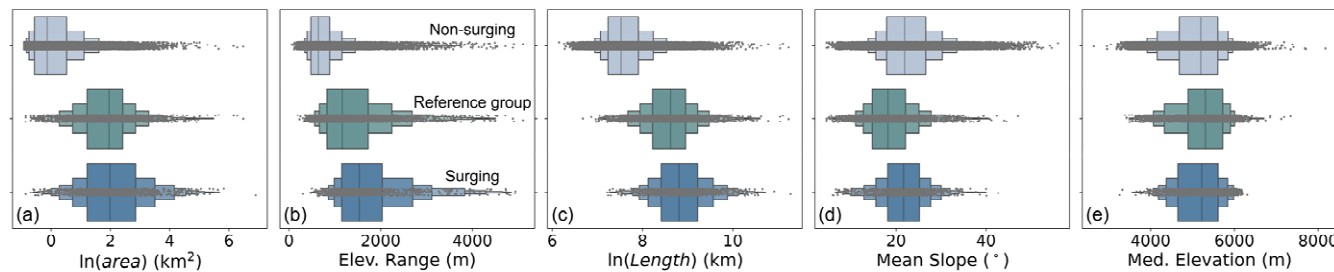

Figure 8: The distribution of glacier number and area in eight aspects. The upper row: glacier number ratio; lower row: glacier area ratio. Left column: distribution of all non-surging glaciers; center column: distribution of non-surging glaciers in the reference group ; right column: distribution of surging glacier. Glaciers smaller than 0.4 km² were excluded in the non-surging glacier class.

Figure 9: The comparison between the boxplots of geometric properties of non-surging glaciers (top), non-surging glaciers in reference group (center) and surging glaciers (bottom). (a) Natural logarithm of area. (b) elevation range. (c) Natural logarithm of length. (d) Mean surface slope. (e) Median elevation. Glaciers smaller than 0.4 km² were excluded in the non-surging glacier class.

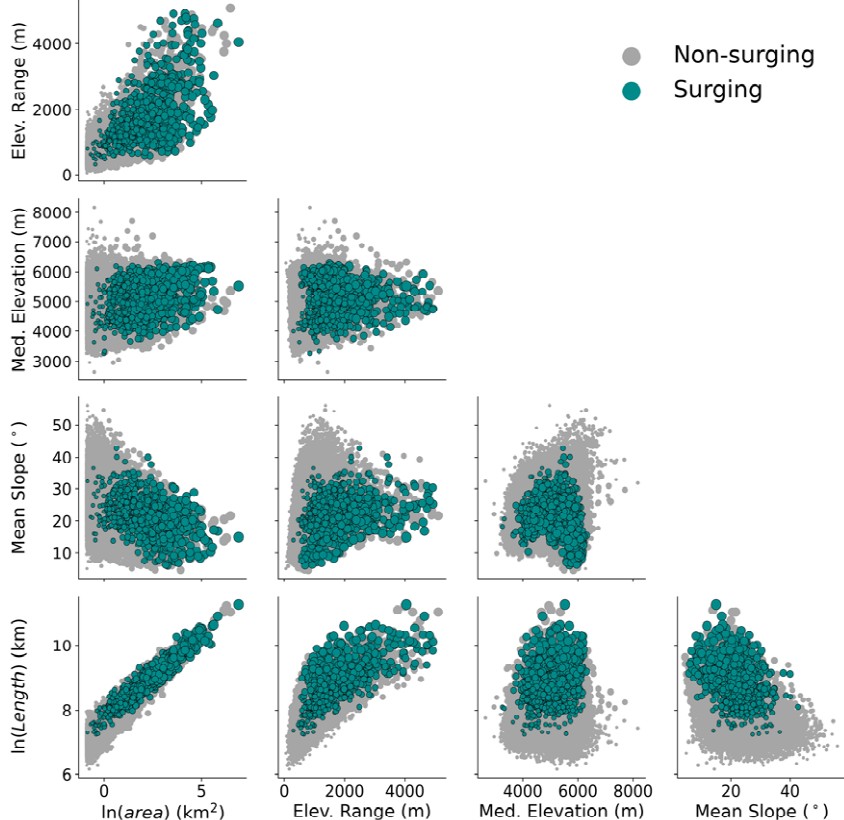

Figure 10: Bivariate scatterplots of geometric properties of non-surging and surging glaciers. The larger dots represent larger glaciers. Glaciers smaller than 0.4 km² were excluded in the non-surging glacier class.

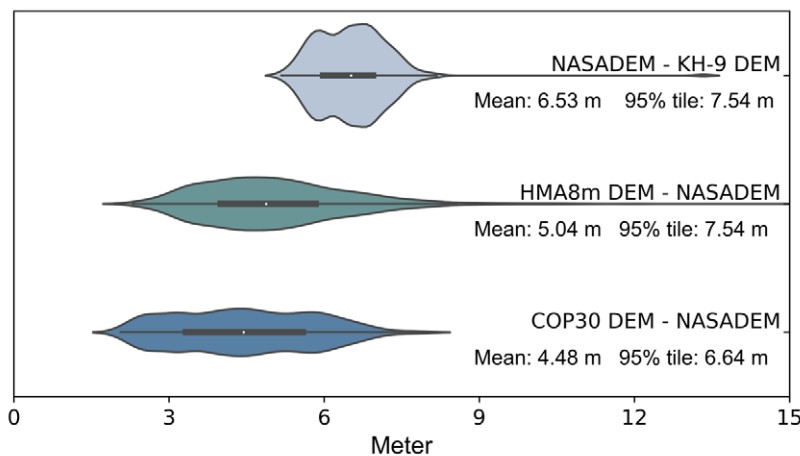

**Figure 11: The distribution of NMAD of elevation change observations in stable areas of all DEM differencing tiles. In each category,**
**the shaded area denotes the density distribution of the NMAD of all DEM differencing tiles. The white dot denotes the median in**
**each group. The thick line represents the interquartile range (IQR, i.e., 75th percentile-25th percentile) in each group. The thin line**
**represents the range between the minimum value (25th percentile - 1.5IQR) and the maximum value (75th percentile + 1.5IQR).**