# Peer review of "A new inventory of High Mountain Asia surging glaciers derived from multiple elevation datasets since the 1970s"

_Earth System Science Data, 2022_

## Referee Comment (RC2)

**General comments**

The authors have investigated glacier elevation changes in High Mountain Asia (HMA) derived from digital elevation models (DEMs) acquired over a period of 50 years to obtain information about surge type glaciers. For this purpose they make use of the usually very typical elevation change pattern that result from the mass transfer of a reservoir zone to the glacier terminus during a surge and the subsequent down-wasting of the tongue and build-up of new mass after a surge. Thereby, the observed elevation changes usually reach several dozens of metres so that the obtained signal is generally much higher than DEM uncertainties. The new assessment is then analysed in terms of topographic glacier characteristics, to extract possible differences between surge-type and other glaciers.

(1) As a general feedback to the terminology, I would say this (and several of the recently published previous studies) presents an inventory of surging glaciers rather than surge-type glaciers. The authors look at real surges that happened rather than investigating surface features (looped moraines etc.) that hint to possible (unobserved) previous surges. This differentiation of terminology should be clearly stated and applied here as it could explain a part of the differences among different inventories.

(2) I think this is about the third or fourth study creating an inventory of surge-type glaciers in HMA, so there is obviously a need for such information. It also seems that it is important which team identified the largest number of 'new' surges. However, as for the previous studies using the RGI6 to mark these glaciers, there is a severe backdrop of the inventory presented here: The authors have not separated the surging glaciers from their trunk glaciers they are connected to in RGI6. This might only be an issue for a smaller portion of glaciers (a few dozens?), but it has to be done. Now the really huge glaciers Baltoro, Biafo, Fedchenko and Siachen are surge-type glaciers although they are not. They are just connected to often much smaller tributaries that have surged. To study these, they have to be separated, maybe just with a straight line to start with (and in the accumulation area). As the authors correctly describe (L325ff), it is clear that in many cases a separation in the ablation region will be difficult, as some surging (tributary) glaciers might contribute substantially to the flow of a trunk glacier or even form a joint tongue with other tributaries. I suggest skipping these complicated cases and focus on the clear ones, e.g. separate Bivachny from Fedchenko, Drenmang and Chiring from Panmah, Maedan from Chiring and Panmah from Choktoi with a straight line.

(3) This separation is not only important for glaciologic studies, but also because the authors analyse here topographic characteristics, size being one of it. When the very large glaciers listed above are in the sample instead of the 10 to 100 times smaller glaciers that really surged, it is not surprising that the result is always that the surge-type glaciers are (on average) larger and longer than the other glaciers. Although this general result might not change after separation, the proper separation is mandatory for a sound assessment of related characteristics. So there is no way around it. A different issue but leading to the same bias is the use of RGI6 for the analysis. There are many smaller glaciers missing in RGI6 and a large number has been manually digitized to the effect that partly huge areas with rock outcrops are included in the glacier outline. This leads to glacier extents that can be too large by 50% or more. Please use for the analysis the outlines provided by the GAMDAM2 inventory (doi.org/10.5194/tc-13-2043-2019).

(4) I had a look at the outlines presented here and compared them to the inventories by Bhambri et al. (2017) for the Karakoram, Goerlich et al. (2020) for the Pamir and Guillet et al. (2022) for entire HMA. Of course, there are differences between the inventories due to different methodological approaches (elevation changes, velocities, length changes), time frame considered or inclusion of surge-type rather than surging glaciers (see above). However, it is argued that the inventory presented here should be more complete as it considers a longer time frame for the analyses. As described in Section 6.3 and confirmed by an overlay of datasets, many glaciers that have been identified as surging in the Guillet inventory are not included here. The authors explain this difference by possibly diluted elevation differences when observed over a longer period of time. I can live with this explanation, but how can this inventory then be more 'complete'? Completeness is mentioned as a key argument for this new survey at many places (L9, 11, 44, 66 and 73). But why should I favour this inventory over the other when obviously so many 'verified' surging glaciers are missing here?

To get the dataset presented here more complete, the authors should at least perform a back check for the glaciers classified as surging in the Guillet et al. inventory but not in this one. A comparison to very high-resolution imagery (e.g. ESRI Basemap) sometimes shows impressively surging glaciers or distorted moraines at these locations. Also a visual inspection of the KH-9 images might reveal that glaciers have surged back then. Some more details about why these glaciers have been missed would be helpful to provide a better insight into the limits of the various approaches. Vice versa, glaciers identified as surge-type here but not by Guillet at al. should also be double-checked. When mass balances are slightly positive and surge marks are missing, it is also possible that glaciers just advance.

(5) Related to the above, the criteria presented for surge identification in Section 4.3 seem to be unbalanced. For example, terminus thickening is included as criterion I-1 but thinning is only in class IV-1, i.e. strong (post-surge) thinning does not have the same weight as thickening. Given that glaciers showing only thickening might just advance rather than surge (creating false positives), I would even argue that strong thinning (see L35) is the better indicator and should at least also be listed in category I. For example, in Figure 2a one can see the strong down-wasting of both Western Kunlun glaciers (just below the last zero of the '2000' annotation) and in Figure 2b the re-advance of its eastern branch, whereas the western branch is still thinning. Indeed, this part is now also surging again and the assignment 'non surge-type' is obviously wrong.

Finally, I suggest removing the climatic interpretation in L342 to 366. We have recently learned that ESSD will refocus on publishing dataset descriptions. This part (although interesting) goes clearly far beyond this. Maybe you can just cite here some studies looking at the interpretation in more detail. Below is a condensed summary of the general points above.

1 Please use proper terminology, distinguishing surging from surge-type
2 Please separate tributaries connected to a larger trunk glacier to the extent possible.
3 Please use GAMDAM2 instead of RGI6
4 Please double-check and correct the present inventory with Gilles et al. and satellite images
5 Please shift class IV-1 'only down-wasting' to class I-4 and re-check then classes I-1 & I-4

**Specific comments**

L35: 'melts fast' this is certainly correct in a relative sense, but still it might take 20 to 30 years or even longer. So maybe write 'relatively fast'.

L41: 'However, a glacier surge' (or use plural: glacier surges are')

L43: When speaking of contemporary glaciers, I suggest writing 'glacier hazards' or maybe even better 'glacier-related hazards'

L44: Glacier-related hazards are certainly an important point for studying surge-type glaciers, but an inventory has no prognostic characteristics and does thus not help to solve the problem. By knowing where these glaciers are, one does not know where the next hazard will occur. I would thus better argue here with the importance of having such an inventory when studying climate change impacts on glaciers, e.g. mass balances or length changes. For such studies it is of high importance to distinguish the two samples.

L67/70: Both studies identified surging glaciers (i.e. surges were observed) rather than surge-type glaciers (which might have surged in the past). Please adjust the terminology.

L73: I would write 'In this study'

L77/78: Such analyses is not allowed in ESSD.

L124: Please use GAMDAM2 instead, RGI6 has quite a lot of issues in this region.

L172: Please also explain in this section how glaciers that are just advancing were distinguished from glaciers that are surging (when criterion I-1 is used).

L182: Please shift the class IV-1 'Strong thinning only' to class I (e.g. as I-4) and get the currently surging west Kunlun Glacier included.

L218: Please revise Section 5.2 after small (part-time) tributary glaciers have been separated from the much larger trunk glaciers. Neither Fedchenko, nor Siachen, Baltoro or Biafo (and several others) are of surge type.

L242: See comment to L218, this applies also to Section 5.3

L296: See comment to L218

L298: This might change a bit when separating small surge-type tributaries from their trunk glaciers. Please get at least the largest ones out of the sample.

L332: Yes, fully agreed, it can be difficult. But it has to be done anyway. It makes no sense to present a topographic analysis for the glacier complex when it is 100 times larger than the glacier that is actually surging.

L356/7: I would not draw such a conclusion when surge cycles are longer than the observation window. This observation can still be by chance.

L396/397: Please clearly separate surge-type glaciers from glaciers with observed surges to get the numbers correctly interpreted.

L400/1: I do not understand this calculation: When Goerlich et al. reports 176 surging (not surge type!) glaciers and the present inventory includes 156 of them, why is this 126 more rather than 20 less? Please better explain what has been calculated and compared here.

L426: Please provide the final selection of glaciers also as point data and in shape file format.

**Figures**

Figs. 2 & 5: The orange to red colours are too close to properly separate them. Please use more different colours.

Fig. 3: The circles can be a it smaller to see more details

Fig. 5: Please revise after glaciers have been separated. Fedchenko, Biafo, Baltoro and Siachen (among many others) are not surge-type glaciers. This is highly misleading information that can not be used for anything.

Figs. 6 and 7: Please use more distinct colours, they are too close.

Fig. 9: Make circles smaller to see something. The foreground data could also be dots only.

---

## Author Comment (AC2)

**Reply to Reviewer #1:**

The manuscript presents a new inventory of surge-type glaciers in High Mountain Asia, derived from glacier surface elevation changes computed from various DEM sources, between the 1970's and 2010's.

The manuscript tackles an important topic which is the identification of surging behavior over a large spatial and temporal scale using remotely sensed glacier observables and thus aims at proposing an updated inventory by incorporating historical data absent from other studies.

This problem is of significant importance or the community and the proposed paper is of overall good quality.

**Major comments:**

1. "The authors here rely solely on anomalous surface elevation change pattern to identify surging behavior. This can lead to false identifications of glaciers present similarly altered surface elevation change signal (See specific comments for more). Some of the widely known shortcomings of the datasets used in this study, as well as the existing corrections (SRTM C-Band penetration correction) are not accounted for in this study, which may lead to further false positive identifications."

Reply: Thanks for reminding. Considering the two error sources you mentioned, we will update the flow chart of data processing and correct the inventory. We believe the newly generated inventory of surging glacier will be more convincible. More details of the revision plan on the false identification handling were listed in the responses for specific comments 13 and 14.

2. "The authors rightfully propose a classification that assigns a level of certainty over the potential surge behavior of each glacier. This is a valid approach since the authors only have one identification criterion and I commend them for doing so. However, when analyzing, discussing and presenting their results, the authors seem to forget that surge-type behavior is uncertain for some glaciers and consistently mention 1015 surge-type glaciers while "only" 704 present indications of surge-type behavior."

Reply: Thanks for reminding. Indeed, speaking of surge-type glacier, only the 'verified' ones should be analyzed and discussed. We will classify the identified glaciers into the "Surge-type" and "Surge-like" categories like Bhambri et al (2017). The uncertain classes ("probable" and "possible") will be classified as "Surge-like" glacier. In the Result section, we will refer to the glacier types more explicitly. Also, only the confirmed surge-type glacier will be taken for characteristic discussion and comparison with previous studies.

3. "Finally, some of the Introduction and Discussion lack context and an adequate description of the state of knowledge of what are glacier surges and the processes that govern glacier instabilities."

Reply: Thanks for reminding. We will elaborate the knowledge of what are glacier surges and the processes that govern glacier instabilities in the Introduction and Discussion.

4. "I want to restate my support for this manuscript and the work it presents. Given its current state however, I

suggest that the authors make major additions and changes to both their methodology and results before this manuscript can be considered for publication.

As an example, adding more than one identification criterion would strengthen the confidence in the presented results. The authors could for example use available satellite images to visually investigate changes in surface and geomorphological features like crevasses, supraglacial ponds or looped moraines."

Reply: Thank you for your affirmation. We will modify our methods and results following your suggestions. In particular, we will add criterions based on morphological feature change. Please see the detailed revision plan for each specific issue listed below.

**Specific comments:**

5. L15-16. This statement is misleading as your classification is based on the confidence degree you have over certain surge events - all the glaciers in your inventory have different level of confidence.

This directly differs from the methodology used in Guillet et al, which I assume is the work you refer to, which used different identification criterion to investigate surges.

Reply: Thanks for reminding. We will categorize the updated results into the "Surge-type" ("verified" class) and "Surge-like" ("probable" and "possible" classes). To avoid confusion, both of them will be clearly stated in the text, especially in the comparison with others results.

6. L23. This relationship was actually first described in the enthalpy balance theory of glacier surges proposed by Benn et al 2019. Please rephrase this statement.

Reply: Thanks for reminding. We would rephrase this statement into "Based on a larger number of samples within HMA, this inventory further confirmed that the glaciers surge activity is more likely to occur for larger and longer glaciers…"

7. L27. Isn't that just because of a sampling bias, as a longer observational period allows to identify more surges?

Reply: Thanks for reminding. We have realized this statement is not rigorous. Since Reviewer #2 also doubt this statement and recommend to remove it, we will remove this statement from our manuscript.

8. L32. Please describe what do those phases entail and how do they differ.

Reply: Thanks for your suggestion. We will add a clear definition of glacier surge to clarify the differences in different surge phase.

9. L34. While I agree that the physics governing the unstable flow exhibited by surge-type glaciers requires a better understanding, I also think that speaking of 'enigma' disregards the substantial efforts made in the recent years to further our understanding of glacier surges.

The authors should here at least refer to the works of Sevestre and Benn (2015), Benn et al (2019), or Thøgersen et al. (2019) (among others) in order to provide an up-to-date synthesis of the state of knowledge on the physics of surge-type glaciers.

Reply: Thanks for reminding. We will rephrase this part and introduce more related works.

10. L35. Please be more specific in this statement. 'Fast' is very vague without a reference. This sentence is not easy to read and could benefit from being segmented and more detailed.

Reply: Thanks for your suggestion. We will rewrite this sentence following your suggestion. We will add more specific number to describe the "unstable flow" and "severe post-surge down-wasting".

11. L46. Not all surging glaciers show terminal advance. I suggest reading through the works of Paul et al. (2017) and Steiner et al. (2018, already cited in your work).
Reply: Thanks for reminding. We will modify the statement to "a surging glacier could exhibit either one or several drastic changes, including…"

12. L58. Contrasting elevation change signal is indeed a powerful tool to identify surge-type glaciers when it is associated to other remotely sensed observations (surface velocity, changes in crevasse pattern etc.). However, the statement made here is misleading as both Lv et al (2019) and Guillet et al (2022) also changes in surface velocity to identify surges. Furthermore, Viay and Braun (2017) focus on the early 21st century - a period of 12 years - not necessarily what could be called a 'long temporal scale'.
Reply: Thanks for reminding. We will alter the references cited here and rephrase the statement. Taking the elevation change as the only criteria to identify surge is indeed not rigorous. We will use more criteria to identify glacier surges following your suggestion (see reply for comment No. 14).

13. L92: As mentioned, the NASADEM used in this study originates from a reprocessing of the C-band SRTM. This data is however known for suffering from important radar penetration. This is more than likely to create spurious elevation change signal in the upper reaches of glaciers which can then lead to false identification of build-up phases, for example. This has to be addressed here, since the NASADEM is extensively used throughout this study.
Reply: Thanks for reminding. We will carefully address the errors caused by radar penetration. We will follow a two-step procedure to reduce the radar penetration bias in the final elevation change results. First, we will difference the NASADEM with the SRTM-X DEM. The elevation differences over glacierized area will be regarded as the penetration difference between X-band and C-band. Secondly, for each 3×3 grid, we will fit a $3^{rd}$ polynomial function between the penetration differences and altitudes. Then, the bias in the glacier dH results (NASADEM minus TanDEM-X DEM) caused by radar penetration will be removed by taking the glacier elevation as input for the function. For the dH results calculated by differencing NASADEM and optical DEMs (HMA8m and KH-9 DEM), the penetration difference between X-band and C- band will be multiplied by 2 to represent the absolute penetration depth of C-band (Fan et al., 2022; Jaber et al.,2019). Also, the uncertainties in the penetration estimation will be included in the analysis of uncertainty of dH.

14. Sec 4.3: This is my main concern with this manuscript. Surge-type glaciers are mainly identified using only surface elevation changes. To me, this approach is a bit hazardous, as many processes can cause altered glacier surface elevation changes compared to what would be deemed as "normal" or "standard". I am here typically thinking about glaciers affected by landslides for example (Hewitt, 2009, Van Wyk de Vries et al., 2022) for example. This needs to be further investigated, including the possibility of using additional criteria to validate the identified surges. Such criteria would typically be morphological (Looped moraine or changes in crevasse patterns) as I doubt of the availability of glacier surface velocity datasets for the 1970s-2000 period.
Reply: Thanks for your suggestion. We have realized that taking the elevation change pattern as the only criteria to identify glacier surges is inadequate. Following your suggestion, we will establish a new workflow to identify surging glacier, which would produce a more reliable inventory. The new workflow is as follow:

1) Firstly, three types of surging glaciers (I-"verified", II-"probable", and III-"possible") will be identified based on

the "abnormal" glacier elevation changes. In particular, glaciers with one of the following four kinds of elevation change features will be identified as "verified" surging glaciers: a) obvious thickening in lower reaches; b) contrasting upper-thinning and lower-thickening; c) contrasting upper-thickening and lower-thinning, which is much stronger than the elevation change caused by normal mass gain/loss; d) severe down-wasting in the lower reaches (as suggested by reviewer 2), which should be either two time stronger than that of the normal glaciers, or be comparable to the strength of adjacent "verified" surging glaciers. Within each observation period, each glacier will be labeled with its possibility level of surging and elevation change pattern (which was missed in the presented inventory).

2) Secondly, long-term Landsat series images (acquired between 1986 and 2021) will be utilized to investigate the morphological change features of the three types of surging glaciers identified from elevation change. For each year, about 2 cloudless Landsat images will be selected. With each Landsat image acquisition frame, images of different dates will be merged into an animated image. Base on the animated image, we are able to identify the morphological feature changes, such as terminus position change, looped moraine changes, and medial moraine changes. Also, we will assign an index of surging possibility for each kind of morphological changes. For example, when the terminus has advanced by over 500m, the glacier will be labeled as "I" type, and as "II" type for less than 500m. Glaciers with clear formation of a new looped moraine or the dramatic shape-change of old looped moraine will be also labeled as "I" type, and those with slight shape-change of old looped moraine will be labeled as "II" type. The treatment of changes in medial moraine is similar to that of looped moraine. The crevasse changes will not be included because the moderate resolution of Landsat images don't allow the identification of crevasse changes.

3) Thirdly, the inventory based on elevation change will be merged with the one based on the morphological change.

4) Finally, we will utilize the very high resolution images (Google/ESRI/Bing, etc.) to check the updated inventory.

This procedure aims to exclude some potential false identifications, such as the severe lower-thinning in a lake-terminated glacier and remarkable surface heightening caused by nearby landslide.

15. L208: Again, this statement is extremely misleading. Glaciers that are considered as "Possibly" or "Probably" surge-type cannot be considered as such. There is no clear surge signal in the elevation change of those glaciers over the studied time periods, otherwise they would be qualified as "verified". In total, you have identified 704 surge-type glaciers at most.

Reply: Thanks for reminding. We will rephrase this sentence. We will clearly differentiate the identified "surge-type" glacier from "surge-like" glaciers in the text, and only refer to the "verified" surge-type glaciers when analyzing the result and making comparison with other studies.

16. L240: Please show examples of these glaciers as well as their elevation change signal for all considered periods.

Reply: We will add the elevation change maps of these identified glaciers for all considered periods.

17. Sec 6.2: This section is very qualitative and provides very few quantitative information. Please provide estimates of the quantities you are referencing.

Reply: Thanks for reminding. We will rewrite this part and incorporate more quantitative information from our results and other studies.

18. L308: You mention randomness, where I believe you mean variability. I do not understand the point made

between L311 and 314. Please clarify.

You further mention that glacier median elevation is "irrelevant" for other topographic parameters. Again this statement is pretty hard to understand, even though I assume you here mean "correlated". A glacier's median elevation is however very correlated to its elevation range. Similarly glacier are is most likely correlated to glacier elevation range (as glaciers are relatively elongated features) so I do not really understand the point of this statement. If I misunderstood your point here please correct me and clarify.

Reply: Yes, we meant the average slope of small glaciers varies widely, and we will modify this statement.

In L311-314 we tried to clarify that, for small glaciers there is no clear relationship between the length and mean slope, because the average slope of small glaciers varies widely. We will modify this statement to make it clearer.

Likewise, we also tried to explain the reason why the distribution of median elevation of surge-type glaciers is similar to non-surge-type glaciers (Fig. 8e). We agree with you that the median elevation is highly correlated to the absolute elevation. However, the elevation range we mentioned means the difference between the maximum and minimum elevation, rather than the absolute maximum and minimum elevation. The median elevation is an absolute value, while the elevation range is a relative value. We will modify this statement to make it clearer.

19. Sec 6.3: This section is a bit problematic at the moment for several reasons. First, the authors propose an inventory based on only one identification criterion and compare it to the one of Guillet et al 2022, which only comprises glaciers for which two criteria of active surges could be observed. Apart from the intrinsic methodological difference, this comparison makes little sense, as the inventory proposed here also comprises glaciers for which surging behavior is uncertain.

This comparison should only target 704 glaciers in the inventory proposed here - which then lowers the difference with Guillet et al (2022) to 38. This in turns leads to several questions, most notably on the length so-called cycle of surging glaciers (assuming that there are no false positives in any of the inventories). Second, the authors mention that with 349 glaciers more (1015-666), they only observe a small (4%) increase in the glacierized area covered by surge-type glaciers in HMA and that, hence, the newly identified ones are relatively small.

This is in stark contrast with previous studies documenting surge-type glaciers as systematically bigger than non-surge type glaciers (Jiskoot et al., 2011, Sevestre and Benn, 2015, to cite a few) as well as what is predicted by the enthalpy balance theory (Benn et al., 2019) and needs further investigation.

Reply: Thanks for reminding. We have realized that the present comparison with other studies is improper. The comparison will be conducted after excluding the uncertainly identified glaciers.

During the past days, we refined our identification strategy by taking your comments into account. We have found some nonnegligible problems in our previous results that were due to the careless check of small glaciers and the underutilization of multi-temporal DEMs (KH-9 and HMA8m). The updated inventory has documented more surge-type glaciers with specific surging features that are mentioned in comment 14. The new inventory should be more reliable. Thanks for your constructive comments. The datasets will be shared soon after the final check and manuscript revision.

References mentioned above:

Bhambri, R., Hewitt, K., Kawishwar, P., and Pratap, B.: Surge-type and surge-modified glaciers in the Karakoram, Scientific Reports, 7, doi:10.1038/s41598-017-15473-8, 2017.

Fan, Y., Ke, C.-Q., Zhou, X., Shen, X., Yu, X., Lhakpa, D., 2022. Glacier mass-balance estimates over High Mountain Asia from 2000 to 2021 based on ICESat-2 and NASADEM. J. Glaciol. 1–13. https://doi.org/10.1017/jog.2022.78

Abdel Jaber, W., Rott, H., Floricioiu, D., Wuite, J., and Miranda, N.: Heterogeneous spatial and temporal pattern of surface elevation change and mass balance of the Patagonian ice fields between 2000 and 2016, The Cryosphere, 13, 2511–2535, doi:10.5194/tc-13-2511-2019, 2019.

---

## Author Comment (AC3)

**Reply to Reviewer2:**

The authors have investigated glacier elevation changes in High Mountain Asia (HMA) derived from digital elevation models (DEMs) acquired over a period of 50 years to obtain information about surge type glaciers. For this purpose they make use of the usually very typical elevation change pattern that result from the mass transfer of a reservoir zone to the glacier terminus during a surge and the subsequent down-wasting of the tongue and build-up of new mass after a surge. Thereby, the observed elevation changes usually reach several dozens of metres so that the obtained signal is generally much higher than DEM uncertainties. The new assessment is then analysed in terms of topographic glacier characteristics, to extract possible differences between surge-type and other glaciers.

**Major comments:**

1. "As a general feedback to the terminology, I would say this (and several of the recently published previous studies) presents an inventory of surging glaciers rather than surge-type glaciers. The authors look at real surges that happened rather than investigating surface features (looped moraines etc.) that hint to possible (unobserved) previous surges. This differentiation of terminology should be clearly stated and applied here as it could explain a part of the differences among different inventories."

Condensed summary: Please use proper terminology, distinguishing surging from surge-type

Reply: Thanks for reminding. We have carefully read the previous studies (Goerlich et al., 2020 and others) and learned the definition difference between surging and surge-type glaciers. We agree with you that our study aiming to present a surging glacier inventory. We will modify all related statement in the text and added a brief definition to distinguish surging glaciers from surge-type glaciers.

2. "I think this is about the third or fourth study creating an inventory of surge-type glaciers in HMA, so there is obviously a need for such information. It also seems that it is important which team identified the largest number of 'new' surges. However, as for the previous studies using the RGI6 to mark these glaciers, there is a severe backdrop of the inventory presented here: The authors have not separated the surging glaciers from their trunk glaciers they are connected to in RGI6. This might only be an issue for a smaller portion of glaciers (a few dozens?), but it has to be done. Now the really huge glaciers Baltoro, Biafo, Fedchenko and Siachen are surge-type glaciers although they are not. They are just connected to often much smaller tributaries that have surged. To study these, they have to be separated, maybe just with a straight line to start with (and in the accumulation area). As the authors correctly describe (L325ff), it is clear that in many cases a separation in the ablation region will be difficult, as some surging (tributary) glaciers might contribute substantially to the flow of a trunk glacier or even form a joint tongue with other tributaries. I suggest skipping these complicated cases and focus on the clear ones, e.g. separate Bivachny from Fedchenko, Drenmang and Chiring from Panmah, Maedan from Chiring and Panmah from Choktoi with a straight line."

Condensed summary: Please separate tributaries connected to a larger trunk glacier to the extent possible.

Reply: Thanks for your suggestion. We have noticed that the tributary or tributary-induced surges would bring biases to the present inventory and result in misleading. To address this problem, we will separate surging tributaries from the stable trunks. A tributary will be considered as a surging glacier if it meets the following criterions. First, the dividing line of contrasting elevation change locates within this tributary. Second, the volume of mass contributed by this tributary to the glacier trunk is relatively small. We have found that many large glaciers are not surging glaciers, and have found about 200 tributary surges in about 100 glacier complexes.

3. This separation is not only important for glaciologic studies, but also because the authors analyse here topographic characteristics, size being one of it. When the very large glaciers listed above are in the sample instead of the 10 to 100 times smaller glaciers that really surged, it is not surprising that the result is always that the surge-type glaciers are (on average) larger and longer than the other glaciers. Although this general result might not change after separation, the proper separation is mandatory for a sound assessment of related characteristics. So there is no way around it. A different issue but leading to the same bias is the use of RGI6 for the analysis. There are many smaller glaciers missing in RGI6 and a large number has been manually digitized to the effect that partly huge areas with rock outcrops are included in the glacier outline. This leads to glacier extents that can be too large by 50% or more. Please use for the analysis the outlines provided by the GAMDAM2 inventory(doi.org/10.5194/tc-13-2043-2019).

Condensed summary: Please use GAMDAM2 instead of RGI6

Reply: Thanks for reminding. The tributary surges will be separated from the glacier complex, and the topographic characteristics of surging glaciers will be reanalyzed based on the new inventory.

We agree with you that the RGI6 inventory has some deficiencies in glacier number and outlines. We will regenerate the surging glacier inventory based on GAMDAM2 inventory from the very beginning of DEM differencing. Since the GAMDAM2 inventory does not includes the geometric and topographic parameters of glaciers, we will firstly calculate the geometric and topographic parameters of each glacier.

4. "I had a look at the outlines presented here and compared them to the inventories by Bhambri et al. (2017) for the Karakoram, Goerlich et al. (2020) for the Pamir and Guillet et al. (2022) for entire HMA. Of course, there are differences between the inventories due to different methodological approaches (elevation changes, velocities, length changes), time frame considered or inclusion of surge-type rather than surging glaciers (see above). However, it is argued that the inventory presented here should be more complete as it considers a longer time frame for the analyses. As described in Section 6.3 and confirmed by an overlay of datasets, many glaciers that have been identified as surging in the Guillet inventory are not included here. The authors explain this difference by possibly diluted elevation differences when observed over a longer period of time. I can live with this explanation, but how can this inventory then be more 'complete'? Completeness is mentioned as a key argument for this new survey at many places (L9, 11, 44, 66 and 73). But why should I favor this inventory over the other when obviously so many 'verified' surging glaciers are missing here?

To get the dataset presented here more complete, the authors should at least perform a back check for the glaciers classified as surging in the Guillet et al. inventory but not in this one. A comparison to very high-resolution imagery (e.g. ESRI Basemap) sometimes shows impressively surging glaciers or distorted moraines at these locations. Also a visual inspection of the KH-9 images might reveal that glaciers have surged back then. Some more details about why these glaciers have been missed would be helpful to provide a better insight into the limits of the various approaches. Vice versa, glaciers identified as surge-type here but not by Guillet at al. should also be double-checked. When mass balances are slightly positive and surge marks are missing, it is also possible that glaciers just advance."

Condensed summary: Please double-check and correct the present inventory with Gilles et al. and satellite images

Reply: Thanks for your suggestion. We will double-check the present inventory with respect to the inventory of Guillet et al. (2022), and update the inventory by introducing the morphological features as an extra criterion. Firstly, the surging behavior will be identified through the multi-temporal and multi-source elevation change maps covering periods of 1970s-2000 and 2000-2020. In this stage, the "surge-like" glaciers will be classified with three level of possibility indicators. After that, we will utilize the long-term Landsat series images (1986-2020) to investigate the morphological feature changes of the "surge-like" glaciers. We will identify the changes of terminus position, looped

moraine, and medial moraine. Also, we will assign a possibility index regarding the strength of morphological feature changes(e.g. "I" for obvious change and "II" for slight change). Then the inventory based on dH will be merged with the one based on morphological change. Finally, we will utilize the very-high-resolution images (Google/ESRI/Bing, etc.) to check the updated inventory. This procedure aims to exclude the false identifications, such as the severe lower-thinning in a lake-terminated glacier, and find more surging glaciers.

5. "Related to the above, the criteria presented for surge identification in Section 4.3 seem to be unbalanced. For example, terminus thickening is included as criterion I-1 but thinning is only in class IV-1, i.e. strong (post-surge) thinning does not have the same weight as thickening. Given that glaciers showing only thickening might just advance rather than surge (creating false positives), I would even argue that strong thinning (see L35) is the better indicator and should at least also be listed in category I. For example, in Figure 2a one can see the strong down-wasting of both Western Kunlun glaciers (just below the last zero of the '2000' annotation) and in Figure 2b the re-advance of its eastern branch, whereas the western branch is still thinning. Indeed, this part is now also surging again and the assignment 'non surge-type' is obviously wrong."

Condensed summary: Please shift class IV-1 'only down-wasting' to class I-4 and re-check then classes I-1 & I-4

Reply: Thank for reminding. We will recalculate the changes in glacier surface elevation and refine the surge identification criterions following your suggestion. The identification criterion based on the elevation change results will be changed into:

I) "verified":

    - a) obvious thickening in lower reaches;

    - b) contrasting upper-thinning and lower-thickening;

    - c) contrasting upper-thickening and lower-thinning, which is much stronger than the elevation change caused by normal mass gain/loss;

    - d) severe thinning in the lower reaches (two time stronger than that of the normal glaciers, or comparable to the ablation of adjacent "verified" surging glaciers);

II) 'probable':

    - a) moderate upper thinning and lower thickening;

    - b) only the moderate thickening in the middle reaches;

III) 'possible':

    - a) only moderate thickening at the terminus;

    - b) only strong thinning in the lower reaches (one time stronger than adjacent normal glaciers);

    Since serious melting can also occur in lake-terminated glaciers, we will remove the lake-terminated glaciers from class "III-b".

6. "Finally, I suggest removing the climatic interpretation in L342 to 366. We have recently learned that ESSD will refocus on publishing dataset descriptions. This part (although interesting) goes clearly far beyond this. Maybe you can just cite here some studies looking at the interpretation in more detail. Below is a condensed summary of the general points above."

Reply: Thanks for reminding. We will delete the climatic interpretation here.

**Specific comments:**

7. L35: 'melts fast' this is certainly correct in a relative sense, but still it might take 20 to 30 years or even longer. So maybe write 'relatively fast'.

Reply: Thanks for reminding. We will rephrase this statement following your suggestion.

8. L41: 'However, a glacier surge' (or use plural: glacier surges are')

Reply: We will rephrase this statement following your suggestion.

9. L43: When speaking of contemporary glaciers, I suggest writing 'glacier hazards' or maybe even better 'glacier-related hazards'

Reply: We will revise the terminology following your suggestion.

10. L44: Glacier-related hazards are certainly an important point for studying surge-type glaciers, but an inventory has no prognostic characteristics and does thus not help to solve the problem. By knowing where these glaciers are, one does not know where the next hazard will occur. I would thus better argue here with the importance of having such an inventory when studying climate change impacts on glaciers, e.g. mass balances or length changes. For such studies it is of high importance to distinguish the two samples.

Reply: Thanks for reminding. We will rewrite this part following your suggestion.

11. L67/70: Both studies identified surging glaciers (i.e. surges were observed) rather than surge type glaciers (which might have surged in the past). Please adjust the terminology.

Reply: Thanks for reminding. We will revise the terminology.

12. L73: I would write 'In this study'

Reply: We will revise this sentence following you suggestion.

13. L77/78: Such analyses are not allowed in ESSD.

Reply: Thanks for reminding. We have noticed the requirement of ESSD publishment. We will remove the unnecessary analysis. We will only take the geometric characteristics for statistically comparison, and will not discuss why they are different.

14. L124: Please use GAMDAM2 instead, RGI6 has quite a lot of issues in this region.

Reply: Thanks for your suggestion. We will use GAMDAM2 instead.

15. L172: Please also explain in this section how glaciers that are just advancing were distinguished from glaciers that are surging (when criterion I-1 is used).

Reply: To take your comment into account, we will establish a new suit of identification criterion which includes the different strength levels of elevation change and morphological feature changes. Please see the replies to comments 4 and 5 for details. Referring to previous studies (Lv et al., 2020; Goerlich et al., 2020), we think the advancing glaciers usually have such features: 1) only thickened in a small area near the glacier terminus, without contrasting upper thinning; 2) the advancing distance is relatively short. These features are corresponding to the "III-a" type of elevation change, and "II" type of terminus advance. Therefore, if a glacier only shows these two kinds of changes, it will be deemed as an advancing glacier, rather than a surging glacier.

16. L182: Please shift the class IV-1 'Strong thinning only' to class I (e.g. as I-4) and get the currently surging west Kunlun Glacier included.

Reply: Thanks for your suggestion. We will refine the surge identification criterion following your suggestion. Please refer to the reply to comment 5 for details.

17. L218: Please revise Section 5.2 after small (part-time) tributary glaciers have been separated from the much larger trunk glaciers. Neither Fedchenko, nor Siachen, Baltoro or Biafo (and several others) are of surge type.

Reply: Thanks for reminding. We have generated an updated surge-type glacier inventory with tributary surges separated from the glacier complex, see reply to comment 2. In the later stage, we will re-conduct the statistical analysis on surging glacier distribution and geometric characteristics difference.

18. L242: See comment to L218, this applies also to Section 5.3

Reply: Thanks for reminding. We will reconduct the analysis as suggested.

19. L296: See comment to L218

Reply: Thanks for reminding. We will reconduct the analysis as suggested.

20. L298: This might change a bit when separating small surge-type tributaries from their trunk glaciers. Please get at least the largest ones out of the sample.

Reply: Thanks for reminding. This part would be revised accordingly during revision, and now we have separated the surging tributaries from the glacier complex with a unified criteria (see reply to comment 2).

21. L332: Yes, fully agreed, it can be difficult. But it has to be done anyway. It makes no sense to present a topographic analysis for the glacier complex when it is 100 times larger than the glacier that is actually surging.

Reply: Thanks for reminding. We have separated the surging tributaries from the glacier complex with a unified criteria in the updated inventory, please see reply to comment 2 for details.

22. L356/7: I would not draw such a conclusion when surge cycles are longer than the observation window. This observation can still be by chance.

Reply: Thanks for reminding. We will remove this conclusion.

23. L396/397: Please clearly separate surge-type glaciers from glaciers with observed surges to get the numbers correctly interpreted.

Reply: Thanks for reminding. We will recheck the Goerlich's inventory and only take the identified surging glaciers for comparison.

24. L400/1: I do not understand this calculation: When Goerlich et al. reports 176 surging (not surge type!) glaciers and the present inventory includes 156 of them, why is this 126 more rather than 20 less? Please better explain what has been calculated and compared here.

Reply: We meant 156 of 176 surging glaciers in Goerlich et al.'s inventory were included by our presented inventory. In Pamir, our inventory recorded 282(200+67+15) surging and surge-type glaciers, and 126 of them were not included by Goerlich et al.'s inventory. However, the inventory will be updated following your and the other reviewer's comments. This comparison will also be revised. We will rephrase these sentences to make them clearer.

25. L426: Please provide the final selection of glaciers also as point data and in shape file format.

Reply: Thanks for reminding. We will provide the identified surging glaciers as point data and in shape file format.

26. Figs. 2 & 5: The orange to red colours are too close to properly separate them. Please use more different colours.

Reply: Thanks for reminding. We will revise these two figures following your suggestion.

27. Fig. 3: The circles can be a bit smaller to see more details

Reply: Thanks for reminding. We will revise Fig. 3 following your suggestion.

28. Fig. 5: Please revise after glaciers have been separated. Fedchenko, Biafo, Baltoro and Siachen (among many others) are not surge-type glaciers. This is highly misleading information that can not be used for anything.

Reply: This figure will be revised according to the updated inventory.

29. Figs. 6 and 7: Please use more distinct colours, they are too close.

Reply: Thanks for reminding. We will use more distinct colors.

30. Fig. 9: Make circles smaller to see something. The foreground data could also be dots only.

Reply: Thanks for reminding. We will revise this figure following your suggestion.

References mentioned above:

Bhambri, R., Hewitt, K., Kawishwar, P., and Pratap, B.: Surge-type and surge-modified glaciers in the Karakoram, Scientific Reports, 7, doi:10.1038/s41598-017-15473-8, 2017.

Goerlich, F., Bolch, T., and Paul, F.: More dynamic than expected: an updated survey of surging glaciers in the Pamir, Earth Syst. Sci. Data, 12, 3161–3176, doi:10.5194/essd-12-3161-2020, 2020.

Lv, M., Guo, H., Yan, J., Wu, K., Liu, G., Lu, X., Ruan, Z., and Yan, S.: Distinguishing Glaciers between Surging and Advancing by Remote Sensing: A Case Study in the Eastern Karakoram, Remote Sensing, 12, 2297, doi:10.3390/rs12142297, 2020.

---

## Author Response (AR1)

**Reply to Reviewer #1:**

The manuscript presents a new inventory of surge-type glaciers in High Mountain Asia, derived from glacier surface elevation changes computed from various DEM sources, between the 1970's and 2010's.

The manuscript tackles an important topic which is the identification of surging behavior over a large spatial and temporal scale using remotely sensed glacier observables and thus aims at proposing an updated inventory by incorporating historical data absent from other studies.

This problem is of significant importance or the community and the proposed paper is of overall good quality.

**Major comments:**

1. "The authors here rely solely on anomalous surface elevation change pattern to identify surging behavior. This can lead to false identifications of glaciers present similarly altered surface elevation change signal (See specific comments for more). Some of the widely known shortcomings of the datasets used in this study, as well as the existing corrections (SRTM C-Band penetration correction) are not accounted for in this study, which may lead to further false positive identifications."

Reply: Thanks for reminding. Considering the two error sources you mentioned, we have updated the flow chart of data processing and correct the inventory. The new flowchart has considered the penetration biases in the multi-source elevation change observations, and incorporated long-term morphological changes as additional identification criteria. More details of the correction of radar penetration and the handling of false identification were described in the responses for specific comments 13 and 14. Specific changes can be found in the Methodology section of the revised manuscript.

2. "The authors rightfully propose a classification that assigns a level of certainty over the potential surge behavior of each glacier. This is a valid approach since the authors only have one identification criterion and I commend them for doing so. However, when analyzing, discussing and presenting their results, the authors seem to forget that surge-type behavior is uncertain for some glaciers and consistently mention 1015 surge-type glaciers while "only" 704 present indications of surge-type behavior."

Reply: Thanks for reminding. Indeed, speaking of surge-type glacier, only the 'verified' ones should be analyzed and discussed. Suggested by Reviewer 2, the verified ones should be referred as 'surging' glaciers, rather than surge-type glaciers. Hence, we have classified the identified glaciers into the "Surging" and "Surge-like" categories like Bhambri et al (2017). The uncertain classes ("probable" and "possible") have been classified as "Surge-like" glacier. In the revised manuscript, only the confirmed surging glacier was taken for characteristic discussion and comparison with previous studies. Specific changes can be found in the Introduction section, 4.3 section, and Results section of the revised manuscript.

3. "Finally, some of the Introduction and Discussion lack context and an adequate description of the state of knowledge of what are glacier surges and the processes that govern glacier instabilities."

Reply: Thanks for reminding. We have elaborated the knowledge of what are glacier surges and the processes that govern glacier instabilities in the Introduction and Discussion sections.

4. "I want to restate my support for this manuscript and the work it presents. Given its current state however, I suggest that the authors make major additions and changes to both their methodology and results before this manuscript can be considered for publication.

As an example, adding more than one identification criterion would strengthen the confidence in the presented results. The authors could for example use available satellite images to visually investigate changes in surface and geomorphological features like crevasses, supraglacial ponds or looped moraines."

Reply: Thank you for your affirmation. We have revised our methods and results following your suggestions. Please see the detailed response for each comment listed below. In particular, we have added the criterions based on morphological feature change to improve the identification of surging glaciers.

**Specific comments:**

5. L15-16. This statement is misleading as your classification is based on the confidence degree you have over certain surge events - all the glaciers in your inventory have different level of confidence.

This directly differs from the methodology used in Guillet et al, which I assume is the work you refer to, which used different identification criterion to investigate surges.

Reply: Thanks for reminding. We have categorized the updated results into the "Surging" ("verified" class) and "Surge-like" ("probable" and "possible" classes). To avoid confusion, both of them were clearly stated in the text, especially in the Abstract, Discussion (comparison with others results), and Conclusion sections.

6. L23. This relationship was actually first described in the enthalpy balance theory of glacier surges proposed by Benn et al 2019. Please rephrase this statement.

Reply: Thanks for reminding. We have rewritten this statement into "The inventory further confirmed that surge activity is more likely to occur for glaciers with larger area, longer length, and wider elevation range…"

7. L27. Isn't that just because of a sampling bias, as a longer observational period allows to identify more surges?

Reply: Thanks for reminding. We have realized this statement is not rigorous. Since Reviewer #2 also doubt this statement and recommend removing it, we have deleted this statement from our manuscript.

8. L32. Please describe what do those phases entail and how do they differ.

Reply: Thanks for your suggestion. We have added a clear definition of different surge phases and described their characteristics in this part.

9. L34. While I agree that the physics governing the unstable flow exhibited by surge-type glaciers requires a better understanding, I also think that speaking of 'enigma' disregards the substantial efforts made in the recent years to further our understanding of glacier surges.

The authors should here at least refer to the works of Sevestre and Benn (2015), Benn et al (2019), or Thøgersen et al. (2019) (among others) in order to provide an up-to-date synthesis of the state of knowledge on the physics of surge-type glaciers.

Reply: Thanks for reminding. We have added a short paragraph to introduce the recent efforts that have been made

10. L35. Please be more specific in this statement. 'Fast' is very vague without a reference. This sentence is not easy to read and could benefit from being segmented and more detailed.

Reply: Thanks for your suggestion. We have rewritten this sentence following your suggestion. We have used more specific number to describe the "unstable flow" and "severe post-surge down-wasting".

11. L46. Not all surging glaciers show terminal advance. I suggest reading through the works of Paul et al. (2017) and Steiner et al. (2018, already cited in your work).

Reply: Thanks for reminding. We have changed the statement to "a surging glacier could exhibit either one or several drastic changes, including…"

12. L58. Contrasting elevation change signal is indeed a powerful tool to identify surge-type glaciers when it is associated to other remotely sensed observations (surface velocity, changes in crevasse pattern etc.). However, the statement made here is misleading as both Lv et al (2019) and Guillet et al (2022) also changes in surface velocity to identify surges. Furthermore, Viay and Braun (2017) focus on the early 21st century - a period of 12 years - not necessarily what could be called a 'long temporal scale'.

Reply: Thanks for reminding. We have altered the references cited here and rephased this part. After adding the additional criteria and refining the results, we have modified this part accordingly to emphasize the significance of using multiple criteria for surging glacier identification.

13. L92: As mentioned, the NASADEM used in this study originates from a reprocessing of the C-band SRTM. This data is however known for suffering from important radar penetration. This is more than likely to create spurious elevation change signal in the upper reaches of glaciers which can then lead to false identification of build-up phases, for example. This has to be addressed here, since the NASADEM is extensively used throughout this study.

Reply: Thanks for reminding. We fully agree with you that the radar penetration could result in nonnegligible biases in the elevation change observations. To address this issue, we followed a two-step procedure to reduce the radar penetration bias in the final elevation change results. First, we differenced the NASADEM with the SRTM-X DEM. The elevation differences over glacierized area will be regarded as the penetration difference between X-band and C-band. Secondly, for each 3×3 grid, we fitted a $3^{rd}$ polynomial function between the penetration differences and altitudes. Then, the bias in the glacier dH results (NASADEM minus TanDEM-X DEM) caused by radar penetration was removed by taking the glacier elevation as input for the function. For the dH results calculated by differencing NASADEM and optical DEMs (HMA8m and KH-9 DEM), the penetration difference between X-band and C- band was multiplied by 2 to represent the absolute penetration depth of C-band (Fan et al., 2022; Jaber et al.,2019). Also, the uncertainties in the penetration estimation were included in the analysis of uncertainty of dH. Specific changes can be found in the 4.1 section of the revised manuscript.

14. Sec 4.3: This is my main concern with this manuscript. Surge-type glaciers are mainly identified using only surface elevation changes. To me, this approach is a bit hazardous, as many processes can cause altered glacier surface elevation changes compared to what would be deemed as "normal" or "standard". I am here typically thinking about glaciers affected by landslides for example (Hewitt, 2009, Van Wyk de Vries et al., 2022) for example. This needs to be further investigated, including the possibility of using additional criteria to validate the identified surges. Such criteria would typically be morphological (Looped moraine or changes in crevasse patterns) as I doubt of the availability of glacier surface velocity datasets for the 1970s-2000 period.

Reply: Thanks for your suggestion. We have realized that taking the elevation change pattern as the only criteria to identify glacier surges is inadequate. Following your suggestion, we have systematically refined the workflow of identifying surging glaciers. The new workflow has taken the long-term morphological changes as an additional criterion. The new workflow is as follow:

1) Firstly, surging glaciers were identified based on the "abnormal" glacier elevation changes. Through the different elevation change pattern, surging glaciers were classified in to three types (I-"verified", II-"probable", and III-"possible"). Within each observation period, each glacier was labeled with its possibility level of surging and elevation change pattern (which was missed in the presented inventory). Specific changes can be found in the 4.3.1 section of the revised manuscript.

2) Secondly, long-term Landsat series images (acquired between 1986 and 2021) were utilized to investigate the morphological change features of the three types of surging glaciers identified from elevation change. The cloudless Landsat images were merged into an animated time-series images, for the visual inspection of morphological feature changes, including terminus position change, looped moraine changes, and medial moraine changes. Also, we have assigned an index of surging possibility (I-"verified", II-"probable") for each kind of morphological changes, according to whether these changes are prominent. The crevasse changes was not included because the moderate resolution of Landsat images didn't allow the identification of crevasse changes. Specific changes can be found in the 4.3.2 section of the revised manuscript.

3) Thirdly, the inventory based on elevation change will be merged with the one based on the morphological change. After that, we performed a careful check on the identification results based on the very high-resolution images (Google/ESRI/Bing, etc.) and compared the newly derived inventory with other inventories. This procedure aims to exclude some potential false identifications, such as the severe lower-thinning in a lake-terminated glacier and remarkable surface heightening caused by nearby landslide. Specific changes can be found in the 4.3.3 section of the revised manuscript.

15. L208: Again, this statement is extremely misleading. Glaciers that are considered as "Possibly" or "Probably" surge-type cannot be considered as such. There is no clear surge signal in the elevation change of those glaciers over the studied time periods, otherwise they would be qualified as "verified". In total, you have identified 704 surge-type glaciers at most.

Reply: Thanks for reminding. We have modified all similar statements. We differentiated the identified "surging" glacier from "surge-like" glaciers in the text, and only refer to the "verified" surging glaciers when analyzing the result and making comparison with other studies.

16. L240: Please show examples of these glaciers as well as their elevation change signal for all considered periods.

Reply: We have added a figure to illustrate the elevation change maps of some identified surging glacier for all considered periods. Please refer to Fig. 6 in the revised manuscript.

17. Sec 6.2: This section is very qualitative and provides very few quantitative information. Please provide estimates of the quantities you are referencing.

Reply: Thanks for reminding. We have rewritten this part and incorporate more quantitative information from our results and other studies.

18. L308: You mention randomness, where I believe you mean variability. I do not understand the point made between L311 and 314. Please clarify.

You further mention that glacier median elevation is "irrelevant" for other topographic parameters. Again this statement is pretty hard to understand, even though I assume you here mean "correlated". A glacier's median elevation is however very correlated to its elevation range. Similarly glacier are is most likely correlated to glacier elevation range (as glaciers are relatively elongated features) so I do not really understand the point of this statement. If I misunderstood your point here please correct me and clarify.

Reply: Yes, we meant the average slope of small glaciers varies widely, and we have modified this statement.

In L311-314 of the original manuscript we tried to clarify that, for small glaciers there is no clear relationship between the length and mean slope, because the average slope of small glaciers varies widely. When analyzing the revised inventory, we set a reference group of non-surging glaciers to minimize the differences between samples of surging and non-surging glaciers. The new analysis result showed that the surging glaciers have a clear higher mean slope. We have rewritten this part according to the new analysis.

Likewise, we also tried to explain the reason why the distribution of median elevation of surging glaciers is similar to non-surging glaciers (Fig. 9e). We agree with you that the median elevation is highly correlated to the absolute elevation. However, the elevation range we mentioned means the difference between the maximum and minimum elevation, rather than the absolute maximum and minimum elevation. The median elevation is an absolute value, while the elevation range is a relative value. We have modified this statement to make it clearer.

19. Sec 6.3: This section is a bit problematic at the moment for several reasons. First, the authors propose an inventory based on only one identification criterion and compare it to the one of Guillet et al 2022, which only comprises glaciers for which two criteria of active surges could be observed. Apart from the intrinsic methodological difference, this comparison makes little sense, as the inventory proposed here also comprises glaciers for which surging behavior is uncertain.

This comparison should only target 704 glaciers in the inventory proposed here - which then lowers the difference with Guillet et al (2022) to 38. This in turns leads to several questions, most notably on the length so-called cycle of surging glaciers (assuming that there are no false positives in any of the inventories). Second, the authors mention that with 349 glaciers more (1015-666), they only observe a small (4%) increase in the glacierized area covered by surge-type glaciers in HMA and that, hence, the newly identified ones are relatively small.

This is in stark contrast with previous studies documenting surge-type glaciers as systematically bigger than non-surge type glaciers (Jiskoot et al., 2011, Sevestre and Benn, 2015, to cite a few) as well as what is predicted by the enthalpy balance theory (Benn et al., 2019) and needs further investigation.

Reply: Thanks for reminding. We have realized that the present comparison with other studies is improper. The comparison has been conducted after excluding the uncertainly identified glaciers.

During the past days, we refined our identification strategy by taking your comments into account. We have found some nonnegligible problems in our previous results that were due to the careless check of small glaciers and the underutilization of multi-temporal DEMs (KH-9 and HMA8m). The updated inventory has documented more surging glaciers with specific surging features that are mentioned in comment 14. The new inventory should be more reliable. Thanks for your constructive comments. The datasets will be shared soon after the final check and manuscript revision.

References mentioned above:

Bhambri, R., Hewitt, K., Kawishwar, P., and Pratap, B.: Surge-type and surge-modified glaciers in the Karakoram, Scientific Reports, 7, doi:10.1038/s41598-017-15473-8, 2017.

Fan, Y., Ke, C.-Q., Zhou, X., Shen, X., Yu, X., Lhakpa, D., 2022. Glacier mass-balance estimates over High Mountain Asia from 2000 to 2021 based on ICESat-2 and NASADEM. J. Glaciol. 1–13. https://doi.org/10.1017/jog.2022.78

Abdel Jaber, W., Rott, H., Floricioiu, D., Wuite, J., and Miranda, N.: Heterogeneous spatial and temporal pattern of surface elevation change and mass balance of the Patagonian ice fields between 2000 and 2016, The Cryosphere, 13, 2511–2535, doi:10.5194/tc-13-2511-2019, 2019.

**Reply to Reviewer2:**

The authors have investigated glacier elevation changes in High Mountain Asia (HMA) derived from digital elevation models (DEMs) acquired over a period of 50 years to obtain information about surge type glaciers. For this purpose they make use of the usually very typical elevation change pattern that result from the mass transfer of a reservoir zone to the glacier terminus during a surge and the subsequent down-wasting of the tongue and build-up of new mass after a surge. Thereby, the observed elevation changes usually reach several dozens of metres so that the obtained signal is generally much higher than DEM uncertainties. The new assessment is then analysed in terms of topographic glacier characteristics, to extract possible differences between surge-type and other glaciers.

**Major comments:**

1. "As a general feedback to the terminology, I would say this (and several of the recently published previous studies) presents an inventory of surging glaciers rather than surge-type glaciers. The authors look at real surges that happened rather than investigating surface features (looped moraines etc.) that hint to possible (unobserved) previous surges. This differentiation of terminology should be clearly stated and applied here as it could explain a part of the differences among different inventories."

Condensed summary: Please use proper terminology, distinguishing surging from surge-type

Reply: Thanks for reminding. We have carefully read the previous studies (Goerlich et al., 2020 and others) and learned the definition difference between surging and surge-type glaciers. We agree with you that our study aiming to present a surging glacier inventory. We have modified all related statement in the text and added a brief definition to distinguish surging glaciers from surge-type glaciers. Also, to avoid similar confusion, we have classified our identified results into the "surging" and "surge-like" groups.

2. "I think this is about the third or fourth study creating an inventory of surge-type glaciers in HMA, so there is obviously a need for such information. It also seems that it is important which team identified the largest number of 'new' surges. However, as for the previous studies using the RGI6 to mark these glaciers, there is a severe backdrop of the inventory presented here: The authors have not separated the surging glaciers from their trunk glaciers they are connected to in RGI6. This might only be an issue for a smaller portion of glaciers (a few dozens?), but it has to be done. Now the really huge glaciers Baltoro, Biafo, Fedchenko and Siachen are surge-type glaciers although they are not. They are just connected to often much smaller tributaries that have surged. To study these, they have to be separated, maybe just with a straight line to start with (and in the accumulation area). As the authors correctly describe (L325ff), it is clear that in many cases a separation in the ablation region will be difficult, as some surging (tributary) glaciers might contribute substantially to the flow of a trunk glacier or even form a joint tongue with other tributaries. I suggest skipping these complicated cases and focus on the clear ones, e.g. separate Bivachny from Fedchenko, Drenmang and Chiring from Panmah, Maedan from Chiring and Panmah from Choktoi with a straight line."

Condensed summary: Please separate tributaries connected to a larger trunk glacier to the extent possible.

Reply: Thanks for your suggestion. We have noticed that the tributary or tributary-induced surges would bring biases to the present inventory and result in misleading. To address this problem, we have separated the surging tributaries from the stable trunks following two criterions: first, the dividing line of contrasting elevation change locates within this tributary; second, the volume of mass contributed by this tributary to the glacier trunk is relatively small. The separation was completed by manual editing. We have found that many large glaciers are not surging glaciers, and

have found 175 tributary surges in 86 glacier complexes. This procedure was completed before the generation of the final inventory.

3. This separation is not only important for glaciologic studies, but also because the authors analyse here topographic characteristics, size being one of it. When the very large glaciers listed above are in the sample instead of the 10 to 100 times smaller glaciers that really surged, it is not surprising that the result is always that the surge-type glaciers are (on average) larger and longer than the other glaciers. Although this general result might not change after separation, the proper separation is mandatory for a sound assessment of related characteristics. So there is no way around it. A different issue but leading to the same bias is the use of RGI6 for the analysis. There are many smaller glaciers missing in RGI6 and a large number has been manually digitized to the effect that partly huge areas with rock outcrops are included in the glacier outline. This leads to glacier extents that can be too large by 50% or more. Please use for the analysis the outlines provided by the GAMDAM2 inventory(doi.org/10.5194/tc-13-2043-2019). Condensed summary: Please use GAMDAM2 instead of RGI6.

Reply: Thanks for reminding. The tributary surges have been separated from the glacier complexes in the updated inventory, and the topographic characteristics of surging glaciers were reanalyzed based on the new inventory.

We agree with you that the RGI6 inventory has some deficiencies in glacier number and outlines. We have recompiled the surging glacier inventory based on GAMDAM2 inventory from the very beginning of DEM differencing. Since the GAMDAM2 inventory does not includes the geometric and topographic parameters of glaciers, we have firstly calculated the geometric and topographic parameters of each glacier.

4. "I had a look at the outlines presented here and compared them to the inventories by Bhambri et al. (2017) for the Karakoram, Goerlich et al. (2020) for the Pamir and Guillet et al. (2022) for entire HMA. Of course, there are differences between the inventories due to different methodological approaches (elevation changes, velocities, length changes), time frame considered or inclusion of surge-type rather than surging glaciers (see above). However, it is argued that the inventory presented here should be more complete as it considers a longer time frame for the analyses. As described in Section 6.3 and confirmed by an overlay of datasets, many glaciers that have been identified as surging in the Guillet inventory are not included here. The authors explain this difference by possibly diluted elevation differences when observed over a longer period of time. I can live with this explanation, but how can this inventory then be more 'complete'? Completeness is mentioned as a key argument for this new survey at many places (L9, 11, 44, 66 and 73). But why should I favor this inventory over the other when obviously so many 'verified' surging glaciers are missing here?

To get the dataset presented here more complete, the authors should at least perform a back check for the glaciers classified as surging in the Guillet et al. inventory but not in this one. A comparison to very high-resolution imagery (e.g. ESRI Basemap) sometimes shows impressively surging glaciers or distorted moraines at these locations. Also a visual inspection of the KH-9 images might reveal that glaciers have surged back then. Some more details about why these glaciers have been missed would be helpful to provide a better insight into the limits of the various approaches. Vice versa, glaciers identified as surge-type here but not by Guillet at al. should also be double-checked. When mass balances are slightly positive and surge marks are missing, it is also possible that glaciers just advance."

Condensed summary: Please double-check and correct the present inventory with Gilles et al. and satellite images

Reply: Thanks for your suggestion. We have refined our workflow for identifying surging glaciers by introducing the morphological feature change as an extra criterion. Firstly, the surging behavior were identified through the multi-temporal and multi-source elevation change maps covering periods of 1970s-2000 and 2000-2020. In this stage, the "surge-like" glaciers were classified with three level of possibility indicators. After that, we utilized the

long-term Landsat series images (1986-2020) to investigate the morphological feature changes of the identified glaciers. We will identify the changes of terminus position, looped moraine, and medial moraine. Also, we will assign a possibility index regarding the strength of morphological feature changes (e.g. "I" for obvious change and "II" for slight change). Then the inventory based on dH will be merged with the one based on morphological change. Finally, we have utilized the very high-resolution images (Google/ESRI/Bing, etc.) to check the updated inventory. This procedure aims to exclude the false identifications, such as the severe lower-thinning in a lake-terminated glacier. We have compared the new inventory with the inventory of Guillet et al. (2022). The differences in the compared inventories were carefully double-checked. For glaciers included in others inventory but not in ours, we performed a back check through our observations, and this added about 20 surging glaciers that were missing by careless. For glaciers in our but not in others inventory, similar operations were conducted with the help of visual inspection on very-high-resolution images. Specific changes can be found in section 4 of the revised manuscript.

5. "Related to the above, the criteria presented for surge identification in Section 4.3 seem to be unbalanced. For example, terminus thickening is included as criterion I-1 but thinning is only in class IV-1, i.e. strong (post-surge) thinning does not have the same weight as thickening. Given that glaciers showing only thickening might just advance rather than surge (creating false positives), I would even argue that strong thinning (see L35) is the better indicator and should at least also be listed in category I. For example, in Figure 2a one can see the strong down-wasting of both Western Kunlun glaciers (just below the last zero of the '2000' annotation) and in Figure 2b the re-advance of its eastern branch, whereas the western branch is still thinning. Indeed, this part is now also surging again and the assignment 'non surge-type' is obviously wrong."

Condensed summary: Please shift class IV-1 'only down-wasting' to class I-4 and re-check then classes I-1 & I-4

Reply: Thank for reminding. We have recalculated the changes in glacier surface elevation and refine the surge identification criterions following your suggestion. The identification criterion based on the elevation change results now have been changed into:

I) "verified":

- a) obvious thickening in lower reaches (e.g. +30 m);

- b) contrasting upper-thinning (e.g. +20 m) and lower-thickening (e.g. +20 m);

- c) contrasting upper-thickening (e.g. +20 m) and lower-thinning (e.g. -30 m);

- d) severe thinning in the lower reaches (two time stronger than that of the normal glaciers, or comparable to the ablation of adjacent "verified" surging glaciers);

II) 'probable':

- a) moderate upper thinning (e.g. -15m) and lower thickening (e.g. +15m);

- b) only the moderate thickening in the middle reaches (e.g. +15m);

III) 'possible':

- a) only moderate thickening at the terminus (e.g. +15m);

- b) only strong thinning in the lower reaches (one time stronger than adjacent normal glaciers).

Since serious melting can also occur in lake-terminated glaciers, we have removed the lake-terminated glaciers from class "III-b".

6. "Finally, I suggest removing the climatic interpretation in L342 to 366. We have recently learned that ESSD will refocus on publishing dataset descriptions. This part (although interesting) goes clearly far beyond this. Maybe you can just cite here some studies looking at the interpretation in more detail. Below is a condensed summary of the general points above."

Reply: Thanks for reminding. We have deleted the climatic interpretation and related discussion here.

**Specific comments:**

7. L35: 'melts fast' this is certainly correct in a relative sense, but still it might take 20 to 30 years or even longer. So maybe write 'relatively fast'.

Reply: Thanks for reminding. We have rephrased this statement following your suggestion.

8. L41: 'However, a glacier surge' (or use plural: glacier surges are')

Reply: We have rephrased this statement following your suggestion.

9. L43: When speaking of contemporary glaciers, I suggest writing 'glacier hazards' or maybe even better 'glacier-related hazards'

Reply: The terminology has been revised following your suggestion.

10. L44: Glacier-related hazards are certainly an important point for studying surge-type glaciers, but an inventory has no prognostic characteristics and does thus not help to solve the problem. By knowing where these glaciers are, one does not know where the next hazard will occur. I would thus better argue here with the importance of having such an inventory when studying climate change impacts on glaciers, e.g. mass balances or length changes. For such studies it is of high importance to distinguish the two samples.

Reply: Thanks for reminding. We have rewritten this part following your suggestion. The revised text has emphasized the significance of surging glacier inventory on the regional mass balance studies, as well as the glacier surging dynamics studies.

11. L67/70: Both studies identified surging glaciers (i.e. surges were observed) rather than surge type glaciers (which might have surged in the past). Please adjust the terminology.

Reply: Thanks for reminding. We have revised the terminology.

12. L73: I would write 'In this study'

Reply: We have revised this sentence following you suggestion.

13. L77/78: Such analyses are not allowed in ESSD.

Reply: Thanks for reminding. We have noticed the requirement of ESSD publishment. We have removed the unnecessary analysis and extended discussions. Only the statistical comparisons between the geometric characteristics of surging and non-surging glaciers were conducted in the revised manuscript.

14. L124: Please use GAMDAM2 instead, RGI6 has quite a lot of issues in this region.

Reply: Thanks for your suggestion. We have replaced the RGI6 with GAMDAM2 in all the processing steps, as well as in the updated inventory.

15. L172: Please also explain in this section how glaciers that are just advancing were distinguished from glaciers that are surging (when criterion I-1 is used).

Reply: Thanks for reminding. As we mentioned above (replies to comments 4 and 5), we have established a new suit of identification criterion which includes the different strength levels of elevation change and morphological feature changes. Referring to previous studies (Lv et al., 2020; Goerlich et al., 2020), we think the advancing glaciers usually have such features: 1) only thickened in a small area near the glacier terminus, without contrasting upper thinning; 2) the advancing distance is relatively short. These features are corresponding to the "III-a" type of elevation change,

and "II" type of terminus advance. Therefore, if a glacier only shows these two kinds of changes, it will be deemed as an advancing glacier, rather than a surging glacier. We have added this part to Section 4.3.3 of the revised manuscript.

16. L182: Please shift the class IV-1 'Strong thinning only' to class I (e.g. as I-4) and get the currently surging west Kunlun Glacier included.

Reply: Thanks for your suggestion. We have refined the surge identification criterion following your suggestion. Specific changes can be found in Section 4.1 of the revised manuscript.

17. L218: Please revise Section 5.2 after small (part-time) tributary glaciers have been separated from the much larger trunk glaciers. Neither Fedchenko, nor Siachen, Baltoro or Biafo (and several others) are of surge type.

Reply: Thanks for reminding. We have generated an updated surging glacier inventory with tributary surges separated from the glacier complex, and we have reconducted the statistical analysis by only taking the 'verified' surging glaciers into account (as suggested by Review 1).

18. L242: See comment to L218, this applies also to Section 5.3

Reply: Thanks for reminding. We have reconducted the analysis as suggested.

19. L296: See comment to L218

Reply: Thanks for reminding. We have reconducted the analysis as suggested.

20. L298: This might change a bit when separating small surge-type tributaries from their trunk glaciers. Please get at least the largest ones out of the sample.

Reply: Thanks for reminding. This part has been revised accordingly during revision. The surging tributaries were separated from the glacier complex with a unified criteria (see reply to comment 2).

21. L332: Yes, fully agreed, it can be difficult. But it has to be done anyway. It makes no sense to present a topographic analysis for the glacier complex when it is 100 times larger than the glacier that is actually surging.

Reply: Thanks for reminding. We have separated the surging tributaries from the glacier complex with a unified criteria in the updated inventory, please see reply to comment 2 for details.

22. L356/7: I would not draw such a conclusion when surge cycles are longer than the observation window. This observation can still be by chance.

Reply: Thanks for reminding. We have removed this conclusion and related discussion.

23. L396/397: Please clearly separate surge-type glaciers from glaciers with observed surges to get the numbers correctly interpreted.

Reply: Thanks for reminding. We have rechecked the Goerlich's inventory and only taken the identified surging glaciers for comparison.

24. L400/1: I do not understand this calculation: When Goerlich et al. reports 176 surging (not surge type!) glaciers and the present inventory includes 156 of them, why is this 126 more rather than 20 less? Please better explain what has been calculated and compared here.

Reply: We meant 156 of 176 surging glaciers in Goerlich et al.'s inventory were included by our presented inventory.

In Pamir, our inventory recorded 282(200+67+15) surging and surge-type glaciers, and 126 of them were not included by Goerlich et al.'s inventory.

However, the inventory has been updated following your and the other reviewer's comments. Now we have reconducted the comparison, and the differences were stated in a clearer way.

25. L426: Please provide the final selection of glaciers also as point data and in shape file format.

Reply: Thanks for reminding. We have uploaded shapefile of the identified surging glaciers as point data.

26. Figs. 2 & 5: The orange to red colours are too close to properly separate them. Please use more different colours.

Reply: Thanks for reminding. We have revised these two figures following your suggestion.

27. Fig. 3: The circles can be a bit smaller to see more details

Reply: Thanks for reminding. We have decreased the circle size and added a stroke to make Fig.3 more readable.

28. Fig. 5: Please revise after glaciers have been separated. Fedchenko, Biafo, Baltoro and Siachen (among many others) are not surge-type glaciers. This is highly misleading information that can not be used for anything.

Reply: This figure has been revised according to the updated inventory.

29. Figs. 6 and 7: Please use more distinct colours, they are too close.

Reply: Thanks for reminding. We have modified both figures with more distinct colors.

30. Fig. 9: Make circles smaller to see something. The foreground data could also be dots only.

Reply: Thanks for reminding. We have revised this figure by reducing the circle size and using more distinct color.

References mentioned above:

Bhambri, R., Hewitt, K., Kawishwar, P., and Pratap, B.: Surge-type and surge-modified glaciers in the Karakoram, Scientific Reports, 7, doi:10.1038/s41598-017-15473-8, 2017.

Goerlich, F., Bolch, T., and Paul, F.: More dynamic than expected: an updated survey of surging glaciers in the Pamir, Earth Syst. Sci. Data, 12, 3161–3176, doi:10.5194/essd-12-3161-2020, 2020.

Lv, M., Guo, H., Yan, J., Wu, K., Liu, G., Lu, X., Ruan, Z., and Yan, S.: Distinguishing Glaciers between Surging and Advancing by Remote Sensing: A Case Study in the Eastern Karakoram, Remote Sensing, 12, 2297, doi:10.3390/rs12142297, 2020.

---

## Referee Report (RR1)

**General comments**

For the revised version of this submission the authors have tried to implement my suggestions to the extent possible, which is highly appreciated. I compared the new PDF to the former one and recognized that large parts of the text have been rewritten, in part also as a result of the improved dataset. In my view the text is now more to the point, the methodology better explained, the categorisation of different surge-types more consistent and the comparison to other datasets completer and better understandable. Of course, the specific thresholds used for the classification can be discussed, but in my view the authors provide a reasonable starting point. The same applies to the combination of the results from two points in time and with other criteria or studies for the *verified*, *probable* and *possible* assignment. Also here the choices of the authors can be discussed, but it is likely impossible to find a 'one fits all' solution. As the authors acknowledge, this study will not be the end of the story, but I think they have done what is currently possible with the datasets available.

I have also checked the revised dataset and think it is much completer and more usable now. The extra effort the authors spent on separating outlines of surging tributaries from often much larger trunk glaciers is highly appreciated. I think this will serve the community much better than currently available datasets and hope this dataset can be integrated in a future update of the new RGI7. Of course, there are now assignments where I would disagree and some glaciers that have obviously surged in the past are not included (e.g. Halong glacier in Tibet described by Wenying (1983) in doi.org/10.3189/S0022143000030306). Hence, also the here presented method (elevation change patterns) has its limitations and further improved and completer results can be obtained when further methods (length and velocity changes) and datasets (literature) are used. As the authors have demonstrated this with a direct comparison to several other datasets, I would leave it with this. The surges that have only be identified in some of these studies are certainly the most interesting, as they will help defining the thresholds needed to distinguish them from other (e.g. just advancing) glaciers.

My most severe objection at this stage is the often wrong English grammar. To get this right, a careful check of the entire text by a native speaker is required. The text is mostly readable, but the wrong grammar is distracting. It seems as if the authors have the usual problems with using 'a', 'the' and the singular or plural form. In many cases they write it just the wrong way round although there are simple rules that can be applied. For example, one should use 'the' when the following is something specific and not use it when it is not (e.g. L113/115 should be 'the COP30 DEM' as the COP30 DEM is something specific and L55 should be 'As for surge-type glaciers, which refers to glaciers that possibly surged ...' as 'surge-type glaciers' are unspecific). But as I am not a native speaker, please give the complete manuscript a check. The points listed below (specific comments) are only a sub-sample. I also suggest to change the wording from 'surging glacier inventory' to 'inventory of surging glaciers' and to use 'HMA DEM' instead of 'HMA8m DEM' and 'NASA DEM' instead of 'NASADEM'.

**Specific comments**

L9: (and elsewhere): '... ice flow and inventories of surging glaciers are important for correctly interpreting regional mass balance'

L10: How can an inventory of surging glaciers help 'assessing glacier-related hazards'? The inventory itself does not indicate anything and even after a hazard has happened, how can the inventory help to assess it? I suggest to remove or explain.

L12: I suggest writing 'In this study …'

L16: I suggest writing 'and 336 probably or possibly surging glaciers'. A 'surge-like glacier' can be something very different.

L17 and elsewhere: I suggest writing 'previous inventories of surging glaciers'

L18: 'excluding glaciers smaller than'

L20: '…uneven. They are …'

L24: have steeper slopes

L26: 'those with severe mass loss'

L39: 'impact the regional mass balance'

L40: investigation and requires to first identify the glacier surges.'

L42/43: insert missing spaces (4 in total)

L444: 'To support related investigations the distribution of surging glaciers is needed as a starting point.

L45: 'studying the internal dynamic processes' For this, also a glacier surge must happen. Just knowing where surges have occurred does not provide anything. I would delete this part or better explain the context.

L45: glacier surges

L48: The Shugar et al. (2021) paper is not related to a surge.

L48: 'A complete inventory of …' Not really. For a hazard assessment you need a hazard. An inventory alone does not help in assessing the hazard nor does it allow to determine where the next hazard will happen.

L51: The 'normal conditions' for a surge-type glacier that is not surging is close to being stagnant, at least its lower parts. This is not a good reference to determine flow acceleration. Maybe refer to the usual flow of glaciers that do not surge.

L52: A medial moraine is common to many glaciers, it is not indicative of surge behaviour. Maybe write 'deformed medial or looped moraines'? As a note, a looped moraine is indicative of a tributary glacier that has surged into a (larger) trunk glacier. In other words, the glacier with the looped moraine is likely not the glacier that surged.

L52: Maybe add 'shear margins' to the list?

L55: As for a surge-type glacier, which refers to a glacier

L58: of supraglacial moraine deformation … To recognize sudden changes

L59: It can also be detected with the change in normalized backscatter from SAR sensors (doi.org/10.5194/tc-15-4901-2021).

L60: The image sources …are limited, the strong changes in glacier motion might be missed.

L61: I think here you mean: 'In contrast, the recognition of a specific surface elevation change pattern is a more reliable way to identify surging glaciers, as it will be visible for many years before and after a surge (…, Zhou et al. 2018). Accordingly, its source datasets (DEMs) can satisfy the required spatio-temporal coverage with … fewer datasets.'

L66: but most effective

L67: 'as a criterion and to combine this information with other observations if possible …'

L70: as well as the differing glacier mass balance

L73: '2020) and Tien Shan'

L74: inventory of surging glaciers

L75: spatio-temporal coverage

L78: '2020), i.e. not all glaciers that surge do also advance.'

L81: long-repetition cycles

L82: surging glaciers

L83: elevation changes

L84: from multiple DEMs

L96: Glacier elevation changes across HMA were found

L98: positive or close to zero changes.

L106: The NASA DEM serves as

L107: with a moderate

L113 of the COP30 DEM

L115: therefore the COP30 DEM

L116: of the product

L119: from very high-resolution

L132: Why 48 m and not 30 or 60?

L134: include the most recent surges (Brun et al., …).'

L138: detected and removed.

L141: 'to capture morphological changes.' I suggest inserting here: 'We acknowledge that due to the 30 m spatial resolution not all details of a changed glacier surface are visible.' Or something similar.

L143: from the USGS

L149: as a template for the inventory of surging glaciers

L151: by also excluding rock outcrops, seasonal snow and shaded

L169: such as the polynomial fit … that were operated

L172: high-mountain regions

L173: 'geolocation shift' The method corrects both horizontal and vertical shifts, which one is meant here?

L189: 'through a Fast-Fourier-Transformation.'

L192: through a 3-sigma threshold criterion.

L195: and potentially lead to false values.

L328: glacier area.

L339: Is it sure that Siachen Glacier surged? In my understanding this is a fast flow glacier such as Baltoro or Biafo that does not really surge.

L349: in the reference group

L352ff: please remove 'the' before north, northwest, north east

L363: in the reference group for comparison,

L365: of the reference group

L368: in the reference group

L371: with glacier area, length and elevation range as these are auto-correlated.

L372: By contrast, glacier median elevation has little correlation with these parameters.

L383: tiles

L390: where surges occur

L422: our results highlight … aspects is slightly

L424/5/9: facing north/towards north/facing north (or 'facing to the north')

L428: north-east facing glaciers have a higher chance to be surging glaciers

L445: positive elevation changes, which is known as one part of the 'Pamir-Karakoram-West Kunlun' Anomaly (as a note: the other two parts are advancing glaciers and stable or even decreasing summer temperatures).

L485: internal glacier surges that did not result in a terminus advance.

L495: observations after 2000 were used

L546: Frank Paul

L729/30: remove capitalization

L759: '(only glaciers larger 0.4 km$^2$ are considered)'

L773: 'The number and area ratios of surging glaciers for different area classes.'

**Figures**

Fig. 1: I suggest making the (a), (b), etc. annotations on the figure a bit larger. Fig. 1c indicates that that the KH-9 coverage is a bit incomplete, in particular in the Tien Shan. Can you add somewhere in the text how many surging or surge-type glaciers have thus no elevation change information for the first period?

Fig. 2: As for Fig. 1, I suggest that you first start with the reference to the figure panel and then describe the contents, i.e. '... change maps during (a) 1970s-2000, (b) 2000-2020 and (c) the corresponding surge classification. The subset selected for visualisation is fine but I miss the link to the classification scheme.' I suggest to annotate panels (a) and (b) with the class derived from the criteria for the glaciers possible. For several of them the assignment to the 'verified' class is not very obvious from the elevation difference maps.

Fig. 3: I suggest using much smaller circles (40% of current size) and more distinct colours to code glacier area (e.g. black, blue, red and yellow). Maybe the sub-regions can be numbered for a quick identification?

Fig. 4: The dark red used to illustrate the percentages is a bit too dark to see the numbers on top. Please use a lighter colour.

Fig. 5: I think the panels are too small to see anything and suggest to show all 4 on top of each other with a width close to page width. I also suggest to use more distinct colours, at least for the probable and possible classes. As mentioned for Fig. 2, please move the panel identifier before the text, i.e. 'identification in (a) the Pamirs, (b) central Tien Shan, ...'

Fig. 6: Maybe use white for the (a) and (b) annotation, it is a bit difficult to see. As for Fig. 2, I suggest to annotate the 'Surging glacier' class in the panel with the result of the classification. When it is not purely the elevation change pattern, please add with an index letter or number (to be explained in the caption) which other criteria have been used.

Fig. 7: Instead of coloured bars you might use two different shades of grey. Please also add minor tick marks on the y-axis and consider using dotted grid lines for the major tick marks of the y-axis. I also suggest to get the inset-table out of the figure and show (and cite) it as a regular table.

Fig. 8: Caption: 'in eight aspect sectors'. As all six panels have an identifier, I suggest repeating it for the column description, e.g. 'Left column (a) and (d): distribution of ...'

Fig. 9: Please add minor tick marks on the x-axis and consider using different shades of grey instead of colour for the plots. Caption: use commas as separators '... area, (b) elevation range, (c) natural logarithm ...'

Fig. 10: I suggest filling the foreground dots with a lighter colour or even white to better see the differently sized circles and to add minor tick marks on all x-axes.

Fig. 11: I suggest using major tick marks every 2 m, insert minor tick marks and dotted major grid lines (or tick marks also at the upper x-axis. Instead of colour, different shades of grey can be used. Please annotate the x-axis with 'NMAD (m)' instead of just 'Meter'.

---

## Author Response (AR2)

**Reply to Editor:**

Dear editor,

Thanks very much for assessing our manuscript. We have carefully read the constructive comments given by you and tried our best to revise the presented manuscript and datasets accordingly. Please see the point-by-point reply for details. We wish to have a chance to send the revised manuscript to you for further consideration. Please do not hesitate to contact us if you have any further comments.

Sincerely,
Lei Guo
On behalf of all co-authors
2 February 2023

**Comments:**

1. Both the reviewers and the editors had requested that your dataset should clearly indicate where the glacier surges occurred. We have noted that you have changed the underlying glacier inventory and that you have trimmed the glacier to the area affected by the surge for further analysis. However, the outlines of the revised dataset is again that of the original dataset without this clear distinction between surge-affected and non-surge affected areas within a given glacier body. We encourage you to also provide your DEM difference products, i.e., those you show in Figures 2 and 6. One solution would be to provide spatially tiled grids (e.g., in 1°x1° grids) of the DEM difference products, or to provide grids for each glacier that you define as having a surge according to their names in the GAMDAM inventory.

Reply: Thanks for your suggestions. We have noticed that both you and reviewers have raised the issue of specifying the location of glacier surges. As Reviewer #2 commented that, we should at least separate the obvious surging tributaries from the very huge glacier complexes. In the presented inventory, we have already done this operation through a self-defined tributary separation strategy, as described in Section 4.3.3. We have also updated the outlines of some surging glaciers with obvious terminus advance through the elevation change map. Therefore, we need to clarify that we have updated the glacier outlines to indicate the surge-affected area in the presented inventory, by trimming the surging tributary and update the terminus position, which is not the original GAMDAM2 outlines anymore. Figure R1 shows the samples of the difference between the glacier outline in our inventory and original GAMDAM2.

However, the delineation of exact location of the active surge is actually too hard to complete. That would lead too much uncertainty as the elevation pattern is indeed a partly and incomplete presence of the surging process. Hope you could understand that.

Following your suggestion, , we have divided the DEM differencing results of surging glaciers into 1×1° tiled grids and uploaded them to the same data repository. Now it should be much easier for others to check our published inventory.

[Figure]

Figure R1: Samples of glacier outline difference in presented inventory and original GAMDAM2 inventory. (a): surging tributary separation (Panmah Glacier); (b) terminus position.

2. In addition, you might consider modifying your point feature dataset so that the point is centered on the glacier surface that had a surge.
Reply: Thanks for reminding. We have manually modified the point-feature inventory. Now the points are centered on the surface of surging glaciers. This dataset has been uploaded to the same data repository.

3. We also noted that in your discussion you could also address the study published by Vale and colleagues last year: Vale, A.B.; Arnold, N.S.; Rees, W.G.; Lea, J.M. Remote Detection of Surge-Related Glacier Terminus Change across High Mountain Asia. Remote Sens. 2021, 13, 1309. https://doi.org/10.3390/rs13071309
Reply: Thanks for reminding. We have added the detailed comparison with this study in the discussion section.

4. Finally, we wondered why you used a subset of the data when comparing glaciers with and without surges (L338-343). Since your analysis focuses primarily on data density and its statistical properties, perhaps you should use the entire data set rather than sampling from the distribution.
Reply: Thanks for your comment. We presented the comparison between all non-surging samples and surging samples in the text, and the subsampled dataset (reference group) was used for additional comparisons to illustrate the biases that could be raised by sampling strategy. In Figures 8 and 9, each comparison contained three groups of glacier samples, including non-surging glaciers (entire dataset), reference group (subsampled non-surging glaciers), and surging glaciers.
We found that the comparison between the surging and all non-surging glaciers is flawed, because the number and distribution of these two groups of samples are quite different. The number of non-surging samples is near 35000, about 35 times of the number of the surging one. Most importantly, among the non-surging glaciers, small ones (<3 km$^2$) take up ~85%, while among surging glaciers the percentage is only 20%. If we considered the very small glaciers (<0.4 km$^2$), this disparity is even larger (~94% vs ~22%). As glaciers of different sizes could show great discrepancy in the geometric attributes, the sample biases would lead to unreliable conclusions of the comparison. Hence, we added comparison with the reference group, among which the sizes of non-surging glaciers are closer to surging glaciers.
Moreover, we found that the comparison between all non-surging glaciers and surging glaciers could

only draw the conclusion that larger glaciers are prone to surge than smaller ones, which has been presented by previous studies. However, it is quite common that two glaciers have similar sizes which only one is surging. So we controlled the area distribution of the non-surging samples to find the potential reasons. We observed that the surging glaciers are generally steeper than the non-surging glaciers that have similar sizes, which has not been reported by previous studies.

**Other changes:**

In order to further improve the quality of the manuscript, we have also done some extra changes after careful check. Most of them are typos or language issues. All changes are highlighted in the updated manuscript.

---

## Author Response (AR3)

**Reply to Reviewer #2:**

**Major comments:**

1. "For the revised version of this submission the authors have tried to implement my suggestions to the extent possible, which is highly appreciated. I compared the new PDF to the former one and recognized that large parts of the text have been rewritten, in part also as a result of the improved dataset. In my view the text is now more to the point, the methodology better explained, the categorization of different surge-types more consistent and the comparison to other datasets completer and better understandable. Of course, the specific thresholds used for the classification can be discussed, but in my view the authors provide a reasonable starting point. The same applies to the combination of the results from two points in time and with other criteria or studies for the verified, probable and possible assignment. Also here the choices of the authors can be discussed, but it is likely impossible to find a 'one fits all' solution. As the authors acknowledge, this study will not be the end of the story, but I think they have done what is currently possible with the datasets available."

Reply: Thanks for your approval of our efforts in revising the manuscript.

2. "I have also checked the revised dataset and think it is much completer and more usable now. The extra effort the authors spent on separating outlines of surging tributaries from often much larger trunk glaciers is highly appreciated. I think this will serve the community much better than currently available datasets and hope this dataset can be integrated in a future update of the new RGI7. Of course, there are now assignments where I would disagree and some glaciers that have obviously surged in the past are not included (e.g. Halong glacier in Tibet described by Wenying (1983) in doi.org/10.3189/S0022143000030306). Hence, also the here presented method (elevation change patterns) has its limitations and further improved and completer results can be obtained when further methods (length and velocity changes) and datasets (literature) are used. As the authors have demonstrated this with a direct comparison to several other datasets, I would leave it with this. The surges that have only be identified in some of these studies are certainly the most interesting, as they will help defining the thresholds needed to distinguish them from other (e.g. just advancing) glaciers."

Reply: Thanks for your encouraging comments on our work. We admit that this inventory does not incorporate all surges during the observation period. Further improvements can be achieved by combining other observations.

3. "My most severe objection at this stage is the often wrong English grammar. To get this right, a careful check of the entire text by a native speaker is required. The text is mostly readable, but the wrong grammar is distracting. It seems as if the authors have the usual problems with using 'a', 'the' and the singular or plural form. In many cases they write it just the wrong way round although there are simple rules that can be applied. For example, one should use 'the' when the following is something specific and not use it when it is not (e.g. L113/115 should be 'the COP30 DEM' as the COP30 DEM is something specific and L55 should be 'As for surge-type glaciers, which refers to glaciers that possibly surged ...' as 'surge-type glaciers' are unspecific). But as I am not a native speaker, please give the complete manuscript a check. The points listed below (specific comments) are only a sub-sample. I also suggest to change the wording from 'surging glacier inventory' to 'inventory of surging glaciers' and to use 'HMA DEM' instead of 'HMA8m DEM' and 'NASA DEM' instead of 'NASADEM'."

Reply: Thanks for reminding. We have employed the English-editing service to improve the language of our manuscript, and have changed the terminologies following your suggestion.

**Specific comments:**

4. L9: (and elsewhere): '… ice flow and inventories of surging glaciers are important for correctly interpreting regional mass balance'

Reply: Thanks for reminding. We have replaced the expression of 'surging glacier inventory' with 'inventory of surging glaciers' throughout the manuscript.

5. L10: How can an inventory of surging glaciers help 'assessing glacier-related hazards'? The inventory itself does not indicate anything and even after a hazard has happened, how can the inventory help to assess it? I suggest to remove or explain.

Reply: Thanks for reminding. We have removed related expressions following your suggestion .

6. L12: I suggest writing 'In this study …'

Reply: Thanks for reminding. We have rewritten it following your suggestion.

7. L16: I suggest writing 'and 336 probably or possibly surging glaciers'. A 'surge-like glacier' can be something very different.

Reply: Thanks for your suggestion. We have rephrased it in the abstract and conclusion sections. In the Results section, we have clearly clarified it to avoid confusion and repetition.

8. L17 and elsewhere: I suggest writing 'previous inventories of surging glaciers'

Reply: We have replaced the expression throughout the entire manuscript following your suggestion .

9. L18: 'excluding glaciers smaller than'

Reply: We have corrected it.

10. L20: '…uneven. They are …'

Reply: We have corrected it.

11. L24: have steeper slopes

Reply: We have corrected it.

12. L26: 'those with severe mass loss'

Reply: We have corrected it.

13. L39: 'impact the regional mass balance'

Reply: We have corrected it.

14. L40: investigation and requires to first identify the glacier surges.'

Reply: We have rephrased this sentence following your suggestion.

15. L42/43: insert missing spaces (4 in total)

Reply: Thanks for reminding. We have corrected it.

16. L44: 'To support related investigations the distribution of surging glaciers is needed as a starting point.

Reply: We have rewritten this sentence following your suggestion.

17. L45: 'studying the internal dynamic processes' For this, also a glacier surge must happen. Just knowing where surges have occurred does not provide anything. I would delete this part or better explain the context.

Reply: Thanks for reminding. We have rephrased this sentence. We wanted to clarify that such inventory is needed to provide various surging glacier samples for studying the internal mechanism of glacier surges.

18. L45: glacier surges

Reply: We have corrected it.

19. The Shugar et al. (2021) paper is not related to a surge.

Reply: Thanks for reminding. We have deleted this citation.

20. L48: 'A complete inventory of …' Not really. For a hazard assessment you need a hazard. An inventory alone does not help in assessing the hazard nor does it allow to determine where the next hazard will happen.

Reply: Thanks for reminding. We have deleted this part.

21. L51: The 'normal conditions' for a surge-type glacier that is not surging is close to being stagnant, at least its lower parts. This is not a good reference to determine flow acceleration. Maybe refer to the usual flow of glaciers that do not surge.

Reply: We have rephrased this part to 'compared to the usual flow of non-surging glaciers'.

22. L52: A medial moraine is common to many glaciers, it is not indicative of surge behaviour. Maybe write 'deformed medial or looped moraines'? As a note, a looped moraine is indicative of a tributary glacier that has surged into a (larger) trunk glacier. In other words, the glacier with the looped moraine is likely not the glacier that surged.

Reply: Thanks for your notes. We have corrected it.

23. L52: Maybe add 'shear margins' to the list?

Reply: Thanks for reminding. We have added it.

24. L55: As for a surge-type glacier, which refers to a glacier

Reply: We have corrected it.

25. L58: of supraglacial moraine deformation … To recognize sudden changes

Reply: Thanks for reminding. We have modified this sentence .

26. L59: It can also be detected with the change in normalized backscatter from SAR sensors (doi.org/10.5194/tc-15-4901-2021).

Reply: Thanks for reminding. We have added the citation of this paper in the text.

27. L60: The image sources …are limited, the strong changes in glacier motion might be missed. The Shugar et al. (2021) paper is not related to a surge.

Reply: Thanks for reminding. We have corrected it.

28. L61: I think here you mean: 'In contrast, the recognition of a specific surface elevation change pattern is a more

reliable way to identify surging glaciers, as it will be visible for many years before and after a surge (…, Zhou et al. 2018). Accordingly, its source datasets (DEMs) can satisfy the required spatio-temporal coverage with … fewer datasets.

Reply: Thanks for reminding. We have rephrased this sentence following your suggestion.

29. L66: but most effective

Reply: We have corrected it.

30. L67: 'as a criterion and to combine this information with other observations if possible …'

Reply: We have corrected it.

31. L70: as well as the differing glacier mass balance.

Reply: We have corrected it.

32. L73: '2020) and Tien Shan'

Reply: We have corrected it.

33. L74: inventory of surging glaciers

Reply: We have corrected it.

34. L75: spatio-temporal coverage

Reply: We have corrected it.

35. L78: '2020), i.e. not all glaciers that surge do also advance.'

Reply: We have added it in the text.

36. L81: long-repetition cycles

Reply: We have corrected it.

37. L82: surging glaciers.

Reply: We have corrected it.

38. L83: elevation changes

Reply: We have corrected it.

39. L84: from multiple DEMs

Reply: We have corrected it.

40. L96: Glacier elevation changes across HMA were found.

Reply: We have corrected it.

41. L98: positive or close to zero changes.

Reply: We have corrected it.

42. L106: The NASA DEM serves as

Reply: We have corrected it.

43. L107: with a moderate

Reply: We have corrected it.

44. L113 of the COP30 DEM

Reply: We have corrected it.

44. L115: therefore the COP30 DEM

Reply: We have corrected it.

45. L116: of the product

Reply: We have corrected it.

46. L119: from very high-resolution

Reply: We have corrected it.

47. L132: Why 48 m and not 30 or 60?

Reply: Processing the KH-9 images covering HMA glaciers is quite time-consuming. To accelerate the computation, a DEM of coarse resolution (48m, with a down-sampling factor of 2) was generated through the stereo processing. Due to the limitations of computation resources, we can only generate the coarser DEMs (lower than 30 m) at this stage. However, the resolution and quality of this DEM should be sufficient for identifying surging glaciers.

48. L134: include the most recent surges (Brun et al., …).'

Reply: We have corrected it.

49. L138: detected and removed.

Reply: We have corrected it.

50. L141: 'to capture morphological changes.' I suggest inserting here: 'We acknowledge that due to the 30 m spatial resolution not all details of a changed glacier surface are visible.' Or something similar.

Reply: Thanks for your suggestion. We have clarified this limitation here.

51. L143: from the USGS

Reply: We have corrected it.

52. L149: as a template for the inventory of surging glaciers

Reply: We have corrected it.

53. L151: by also excluding rock outcrops, seasonal snow and shaded

Reply: Thanks for reminding. We have modified this sentence.

54. L169: such as the polynomial fit … that were operated

Reply: We have corrected it.

55. L172: high-mountain regions

Reply: We have corrected it.

56. L173: 'geolocation shift' The method corrects both horizontal and vertical shifts, which one is meant here?

Reply: Both kinds of shifts were corrected through this method. We have clarified it.

57. L189: 'through a Fast-Fourier-Transformation.'

Reply: We have corrected it.

58. L192: through a 3-sigma threshold criterion.

Reply: We have corrected it.

59. L195: and potentially lead to false values.

Reply: We have corrected it..

60. L328: glacier area.

Reply: We have corrected it.

61. L339: Is it sure that Siachen Glacier surged? In my understanding this is a fast flow glacier such as Baltoro or Biafo that does not really surge.

Reply: Thanks for reminding. In the last round of revision, we carefully checked the identification results of huge glaciers. In periods of 1974-2000 and 1980-2000, the Siachen glacier has showed obvious thickening in the tongue (up to 50 m), and slight thinning in the upper reaches. We thought this could be the signal at the late stage of a previous surge, or possibly a mini surge. According to our identification criteria, we finally decided to classify it as surging.

[Figure]

Fig.1 Elevation changes of the Siachen glacier during periods 1974-2000 (left) and 1980-2000 (right).

62. L349, L363, L365, L368: in the reference group

Reply: We have corrected them.

63. L352ff: please remove 'the' before north, northwest, north east

Reply: We have corrected it.

64. L371: with glacier area, length and elevation range as these are auto-correlated.

Reply: We have corrected it.

65. L372: By contrast, glacier median elevation has little correlation with these parameters.

Reply: We have corrected it.

66. L383: tiles

Reply: We have corrected it.

67. L390: where surges occur

Reply: We have corrected it.

68. L422: our results highlight … aspects is slightly

Reply: Thanks for reminding. We have corrected it.

69. L424/5/9: facing north/towards north/facing north (or 'facing to the north')

Reply: We have corrected it.

70. L428: north-east facing glaciers have a higher chance to be surging glaciers

Reply: We have corrected it.

71. L445: positive elevation changes, which is known as one part of the 'Pamir-Karakoram-West Kunlun' Anomaly (as a note: the other two parts are advancing glaciers and stable or even decreasing summer temperatures).

Reply: Thanks for your note. We have corrected it.

72. L485: internal glacier surges that did not result in a terminus advance.

Reply: We have corrected it.

73. L495: observations after 2000 were used

Reply: This study has utilized multiple elevation change observations covering the periods both before and after 2000. Here we are trying to say that the elevation change observations before 2000 used by them only cover a small part of the Western Pamir, which could lead to fewer identified surges before 2000. We have rewritten this sentence to make it clear.

74. Frank Paul

Reply: We are sorry about the carelessness. We have corrected it.

75. L729/30: remove capitalization

Reply: We have corrected it.

76. L759: '(only glaciers larger 0.4 km2 are considered)'

Reply: We have corrected it.

77. L773: 'The number and area ratios of surging glaciers for different area classes.'

Reply: We have corrected it.

**Figures:**

78. Fig. 1: I suggest making the (a), (b), etc. annotations on the figure a bit larger. Fig. 1c indicates that the KH-9 coverage is a bit incomplete, in particular in the Tien Shan. Can you add somewhere in the text how many surging or surge-type glaciers have thus no elevation change information for the first period?

Reply: Thanks for your suggestions. We have modified the label size in Fig.1. The incomplete coverage of usable KH-9 images led to 103 surging glaciers having no observations during the period of 1970s-2000. We have clarified this at the beginning of the results section.

79. Fig. 2: As for Fig. 1, I suggest that you first start with the reference to the figure panel and then describe the contents, i.e. '… change maps during (a) 1970s-2000, (b) 2000-2020 and (c) the corresponding surge classification. The subset selected for visualisation is fine but I miss the link to the classification scheme.' I suggest to annotate panels (a) and (b) with the class derived from the criteria for the glaciers possible. For several of them the assignment to the 'verified' class is not very obvious from the elevation difference maps.

Reply: We have added annotation to panels (a) and (b) with the classes derived from the identification criteria and modified the caption here. This study has utilized multiple elevation change observations covering the periods both before and after 2000. However, only one elevation change observation during periods both before and after 2000 was illustrated in this figure, and therefore the final identified class may not correspond to the elevation change pattern. In view of this issue, we have also added a subscript to note that elevation change observations of other periods were combined to determine the surging class.

80. Fig. 3: I suggest using much smaller circles (40% of current size) and more distinct colours to code glacier area (e.g. black, blue, red and yellow). Maybe the sub-regions can be numbered for a quick identification?

Reply: Thanks for reminding. We have modified the circle size and color following your suggestion. We have added the names of sub-regions in the figure.

81. Fig. 4: The dark red used to illustrate the percentages is a bit too dark to see the numbers on top. Please use a lighter colour.

Reply: We have changed the dark red to a lighter color following your suggestion.

82. Fig. 5: I think the panels are too small to see anything and suggest to show all 4 on top of each other with a width close to page width. I also suggest to use more distinct colours, at least for the probable and possible classes. As mentioned for Fig. 2, please move the panel identifier before the text, i.e. 'identification in (a) the Pamirs, (b) central Tien Shan, …'

Reply: The size of this figure has been increased. We also changed the color and modified the caption following your suggestion.

83. Fig. 6: Maybe use white for the (a) and (b) annotation, it is a bit difficult to see. As for Fig. 2, I suggest to annotate the 'Surging glacier' class in the panel with the result of the classification. When it is not purely the elevation change

pattern, please add with an index letter or number (to be explained in the caption) which other criteria have been used.

Reply: After many times of test, we found that adding a background to the annotations is a better way to make them clearer. We have added the annotations of the surging class following the way in Fig. 2. Thanks for reminding.

84. Fig. 7: Instead of coloured bars you might use two different shades of grey. Please also add minor tick marks on the y-axis and consider using dotted grid lines for the major tick marks of the y-axis. I also suggest to get the inset-table out of the figure and show (and cite) it as a regular table.

Reply: We have modified this figure following your suggestion. The inset table has been taken out and turned to Table 3 in the revised manuscript.

85. Fig. 8: Caption: 'in eight aspect sectors'. As all six panels have an identifier, I suggest repeating it for the column description, e.g. 'Left column (a) and (d): distribution of …'

Reply: We have modified the caption following your suggestion.

86. Fig. 9: Please add minor tick marks on the x-axis and consider using different shades of grey instead of colour for the plots. Caption: use commas as separators '… area, (b) elevation range, (c) natural logarithm …'

Reply: We have modified the figure and caption following your suggestion.

87. Fig. 10: I suggest filling the foreground dots with a lighter colour or even white to better see the differently sized circles and to add minor tick marks on all x-axes.

Reply: We have modified the color and added the minor ticks following your suggestion.

88. Fig. 11: I suggest using major tick marks every 2 m, insert minor tick marks and dotted major grid lines (or tick marks also at the upper x-axis. Instead of colour, different shades of grey can be used. Please annotate the x-axis with 'NMAD (m)' instead of just 'Meter'.

Reply: We have modified this figure following your suggestion .

References mentioned above:

Dehecq, A., Gardner, A. S., Alexandrov, O., McMichael, S., Hugonnet, R., Shean, D., and Marty, M.: Automated Processing of Declassified KH-9 Hexagon Satellite Images for Global Elevation Change Analysis Since the 1970s, Front. Earth Sci., 8, 566802, doi:10.3389/feart.2020.566802, 2020.

**Reply to Reviewer #3:**

**Specific Comments:**

1. L37: Please refer to the specific inventory you have compared?

Reply: Thanks for reminding. We have rephrased it to clarify that the number was concluded based on the comparison with the most recent study, which refers to the inventory presented by Guillet et al. (2022).

2. L19-L22: The regions listed in the bracket have been directly described in the result and discussion sections. So, you can remove these sentences to keep the abstract brief.

Reply: Thanks for reminding. We have modified it as you suggested.

3. L44: Replace "accurate" with "direct" or "clear"? Because the "distribution" cannot be quantitatively described.

Reply: Thanks for reminding. We have rephrased this sentence to 'the distribution of surging glaciers is needed as a starting point'.

4. L56: I would say that less calculation does not mean the visual interpretation of glacier surface morphological changes is "easy to operate". Please rephrase it.

Reply: Thanks for reminding. We have modified this expression to 'less calculative'.

5. L85: "The preliminary identified surging glaciers were divided into…".

Reply: Thanks for suggesting. We have modified it.

6. L130: There should be a comma after "Alaska".

Reply: We have corrected it.

7. L133: Put the "(See Fig. 1c)" at the end of previous sentence.

Reply: Thanks for reminding. We have modified it.

8. L212: "were" → "was".

Reply: We have corrected it.

9. L215: Delete the redundant blank between "the" and "false".

Reply: We have corrected it.

10. L259: "small" → "slight".

Reply: We have modified it.

11. L299: "all" → "most"? In fact, your elevation change results do not have 100% coverage, especially for the period of 1970s.

Reply: Thanks for reminding. We have modified it to 'most'. Besides, we have added a sentence to demonstrate the coverage of KH-9 DEMs here.

12. L381: I would say the NMAD "can be deemed as a better proxy of uncertainties in dH than STD".

Reply: Thanks for your suggestion. We have rephrased it.

13. L390: "top" → "head"?
Reply: We have corrected it.

14. L453: The expression of "809 if RGI6.0 was used" is ambiguous. Maybe you can rewrite it to "809 if represented by RGI6.0 polygons"?
Reply: Thanks for reminding. We have modified the expression following your suggestion.

15. L767: Figure 6: if the surging glaciers were identified through multiple criteria and multi-temporal changes, then I can understand that why some glaciers (e.g. the largest one in panel a) are classified as surging. But you should clarify that more clearly in the caption text.
Besides, please use a thinner line of the surging glaciers to make this figure more readable."
Reply: Fig.6 has been redrawn with a thinner line to indicate identified surging glaciers. After that, we also added a label to annotate the specific identified class of the surging glaciers. Since only one frame of elevation change during each observation period was illustrated in this figure, the final identified class may not correspond to the elevation change pattern. We have also added two subscripts to demonstrate that dH of other periods or morphological changes were combined to determine the surging class, respectively. We have added the explanation in the caption.

References mentioned above:
Guillet, G., King, O., Lv, M., Ghuffar, S., Benn, D., Quincey, D., and Bolch, T.: A regionally resolved inventory of High Mountain Asia surge-type glaciers, derived from a multi-factor remote sensing approach, The Cryosphere, 16, 603–623, doi:10.5194/tc-16-603-2022, 2022.

---

## Author Response (AR4)

**Reply to editors:**

**Major Comments:**

1. "One point is some remaining problems in language and terminology. We hope that you will find our suggestions in the attached PDF helpful and ask that you further revise your manuscript again on these technical details."

Reply: Thanks for providing the detailed suggestions on the technical issues. We have revised the manuscript and modified the text according to your comments. In addition, we have also carefully revised the language and expression throughout the manuscript. Some repetitive sentences have been deleted (Lines 218-220, which are similar to Lines 257-259). You can find our changes in the uploaded manuscript with change marks.

2. "In addition, we have discussed in the editorial team that uncertainty analysis should be key to any publication in the ESSD. To this end, we ask that you quantify the sources of uncertainty more precisely and present the error propagation in more detail. Further advise can be found in the attached document in section 6.1."

Reply: Thanks for reminding. We have reevaluated the uncertainties of glacier change following your suggestions. We have added a section in the method part to show the details of how the uncertainty is calculated through the error propagation law. The contributions of different uncertainty sources were quantified separately. We have added a subplot in Figure 1o to show the contributions of different uncertainty sources. Besides, we have added more quantitative supports and qualitive discussion to demonstrate that the uncertainties in our results is acceptable. The uncertainty analysis section (6.1) has been significantly revised. Please find our detailed revision for each related comments in the replies to comments 23-27.

3. "Finally, we noticed that the presentation and discussion of the characteristics of glacier surges takes up a lot of space in the manuscript. However, a scientific discussion of your database, apart from methodological issues and comparison with other studies in terms of completeness, is discouraged by the ESSD guidelines. We note that we have not raised this issue before, but would at least ask you to consider shortening or deleting the relevant sections so that your manuscript is more in line with the ESSD guidelines."

Reply: Thanks for reminding. We have carefully learned the guidelines for ESSD publications. Following your suggestion, we have deleted the content related to glacier orientation in the Results section (section 5.3), and the content related to glacier orientation and mass balance in the Discussion section (section 6.2). The content related to geometric characteristics in the Discussion section (section 6.2) has also been significantly shortened. The revised manuscript should be more in line with ESSD guidelines now.

**Specific Comments:**

4. L32: remove "quasi-".

Reply: A surge is not exactly periodic, since its revisit cycle is affected by the environmental conditions. For periodic glacier surges the environmental conditions would need to remain stable. Glacier surge was defined as a "quasi-periodic" behavior in many studies (Raymond, 1987; Harrison and Post, 2003; Kochtiziky et al., 2019; Truffer et al., 2021). Hence, the "quasi-" should be kept.

4. L33: replace with "the accumulation area"?

Reply: Thanks for reminding. We have replaced the "upper reaches" with "the reservoir zone". The reservoir zone of a surging glacier is not necessarily in the accumulation area.

5. L46: "distinct elevation change pattern" should be more specific

Reply: Thanks for reminding. We have rephrased this expression and added the typical elevation change pattern for identifying a surging glacier.

6. L60: "By combining observation of multiple features" is unclear what that means. Please specify.

Reply: Thanks for reminding. We have rewritten this sentence to specify the meaning.

7. L66: "differing" means differing between or within regions?

Reply: it means the mass balance of these subregions is different from that of other regions outside HMA. We have rephrased this expression.

8. L130: add period for "the most recent surges"

Reply: Thanks for reminding. We have added the corresponding period for this expression.

9. L133: "the elevation change pattern" in which data set? Only the datasets by brun and hugonnet?

Reply: We meant the errors in all elevation change observations, including our DEM differencing results and existing elevation change datasets. We have clarified that.

10. L158: which resampling algorithm did you used for reprojection?

Reply: We used the cubic resampling method for map reprojection. We have added a sentence to clarify it.

11. L159: did you use an Albers projection specific for Asia?

Reply: This map projection is a customed Albers projection to encompass all glaciers in HMA, which was used by Shean et al., (2020). We have clarified it.

12. L195: explain "dH"

Reply: Thanks for reminding. We have added the explanation when we used it at the first time. It was used for indicating elevation difference.

13. L198: why this multiplication?

Reply: We used this multiplication according to previous studies (Abdel Jaber et al., 2019; Fan et al., 2022). Also, the experiments conducted over dry snow showed that the penetration depth of C-band is about two times of X-band (table 2 in Rott H et al., 1993). We have rephrased this sentence and added the references in the text.

14. L199: what do you mean with "removed"?

Reply: Thanks for reminding. We have replaced "removed" with "subtracted".

15. L225: do these values refer to individual pixels or larger areas of a given size? please clarify

Reply: Thanks for reminding. These values refer to thresholds of continuous glacier elevation changes over an area larger than 0.04 km$^2$, rather than individual pixels. We have clarified that in the text.

16. L271: the logic and use behind the Roman numbers I, II, and III is a bit confusing, because they are used both in chapters 4.2.1 and 4.2.2. Also the letters a, b, and c are not clearly defined. Please add a table that clarify this issue, and remove/ relabel all phrases in the text where potential confusion between these numbers and letter could arise.

Reply: Thanks for reminding. We have revised all related phrases to avoid confusion in both the text and dataset. Also, we have added a table to list all criteria used in this study (Table 1).

17. L303: spatial or temporal "density"?

Reply: We mean spatial density. We have clarified it.

18. L304: "far from even" between what? Regions?

Reply: we mean the spatial density of identified surging glaciers varies between different subregions. We have rephrased this expression.

19. L315: does this estimate include the entire area of the glacier, including that part that surged, and the other part that did not surge? Please clarify.

Reply: Only the area of surging tributaries was considered in all area-related values. The other part that did not surge was not counted. We have clarified that a surging tributary was regarded as an individual glacier in the previous paragraph.

20. L335: could that drop be the result of the overall small number of glaciers in that bin?

Reply: Yes, it could be the reason. However, we cannot say this view as the samples are insufficient.

21. L340: "closest area" but from an arbitrary region?

Reply: Yes, it is. Thanks for reminding. We have added these words.

22. L372: Please add here a table of all variables and their units that are part of the shapefile that are avaiable on Zenodo. Thanks.

Reply: Thanks for your careful review. We have listed the description of all variables of the shapefiles that are available on Zenodo in Table 5. We have added a sentence here to clarify it.

23. L374: Please revisit your workflow and quantify the different contributions of uncertainty until you obtain your output product (i.e., the DEM difference maps)

Reply: We have reevaluated the uncertainties of glacier change following your suggestions. The uncertainties of the elevation difference and penetration depth difference were separately evaluated prior to the final estimation of uncertainties. Regarding this, we have added a subplot in Figure 10 to show the uncertainties of all kinds of elevation difference observations. We have significantly revised sections 4.3 and 6.1.

24. L378: which you did not show in the methods. Please add in the methods how you propagate errors of DEMs, and in the DEM difference products. Please also discuss how you much the signal of elevation change must be to be credibly higher than the noise in your difference product.

Reply: We have added a section in 'Method' (section 4.3) to clarify how we estimated the uncertainties in the elevation change through the error propagation law. We have added discussion on the form and performance of large errors in our elevation difference results, and the reason why these large errors would not substantially affect our identification in section 6.1.

25. L383: how do you know that these values are uncertainties and not 'true' deviations? Please show/ discuss.

Reply: It could be that in our "stable terrain" there are other processes occurring, like landslides, erosion, anthropogenic changes etc. The uncertainty is calculated by taking the NMAD of elevation difference values in "stable terrain" where glaciers and water bodies are excluded. It is possible that some of the differences are actual signal (landslides, erosion, vegetation change etc), but these are likely to be in minority over the whole study area. In any case, our error estimate is likely conservative, since it may overestimate the actual uncertainty.

26. L391: please be more specific.

Reply: We have added a sentence to clarify this point.

27. L388: please add quantitative support to this claim.

Reply: We have added a figure to illustrate the relationship between uncertainties and terrain slope (Fig.11), which shows that higher uncertainties are more likely to occur in steep regions. The head of glaciers are generally steep regions. We have added related discussion in section 6.1.

28. L399: what is a glacier scale?
Reply: Thanks for reminding. We have replaced the "scale" with "size".

29. L467: which study used only a single criterion? Didn't Guillet's and your study used several criteria to examine glacier surges? Please clarify.
Reply: Sorry for the carelessness. This expression actually corresponds to the first version of our manuscript, in which we have not added the morphological changes as the criteria. We have corrected it.

30. L478: "more surging glaciers" in the Karakoram?
Reply: Thanks for reminding. We have added "in the Karakoram" in the text.

31.L485: why did you miss these four cases?
Reply: We have rechecked these four glaciers, and found that all of them showed very slight terminus advancing and their surging lasted very long time, which means their terminus advancing speeds were very low (i.e., lower than 1 pixel of the Landsat image per year). Therefore, it is difficult to identify these surges through the visual interpretation. We have added the reason in the text.

32. Landsat has higher spatial resolution than some of your DEMs?
Reply: Thanks for reminding, we should modify this expression. However, we have realized that the discussion of the reason for the gap between the numbers of identified surging glaciers of Vale et al (2021) and us is unnecessary, because they only used the terminus change as the criterion. We decided to delete these two sentences (i.e., "The possible reason …lacier terminus position).

33. Figure 2: These labels I-b, I-c, and so on, need to be explained in the figure caption. Please add a small overview map where this site is.
Reply: We have added the overview map in the figure. Also, the labels for identified surging glaciers were modified according to comment 16.
We understand that it is convenient for readers to quickly know what the labels stand for after we add the explanation in the figure caption. We have tried to this. Here is the explanation of these labels:
"I-a": a "verified" surging glacier that was observed to have obvious thickening in lower reaches (e.g., +30 m); "I-b": a "verified" surging glacier that was observed to have contrasting upper-thinning (e.g.,+20 m) and lower-thickening (e.g.., -20 m); "I-c": a "verified" surging glacier that was observed to have contrasting upper-thickening (e.g.., +20 m) and lower-thinning; "I-d": a "verified" surging glacier that was observed to have severe thinning in the lower reaches;
"II-e": a "probable" surging glacier that was observed to have moderate upper thinning (e.g., -15m) and lower thickening (e.g., +15 m);
"II-f": a "probable" surging glacier that was observed to have only moderate thickening in the middle reaches (e.g., +15 m);
"III-g": a "possible" surging glacier that was observed to have only moderate thickening at the terminus (e.g., +15 m); "III-h": a "possible" surging glacier that was observed to have only strong thinning in the lower reaches.
The explanation of these labels takes up about two hundreds of words, making the caption lengthy. How about we point out that the criteria of identification are elaborated in section 4.2.1 and Table 1? It will be much more concise.

34. Figure 4: add this label to the figure, e.g. in the legend in the topright, or the estimate for the entire HMA in the bottom left.
Reply: Thanks for reminding. We have added the label into the figure.

35. Figure 10: could transparency for both types of glaciers help reduce the strong overlap? Stretch labels.

Reply: We have modified this figure following your suggestion. However, it's difficult to clearly show both type of glaciers due to the large number of samples.

References mentioned in the text:

Abdel Jaber, W., Rott, H., Floricioiu, D., Wuite, J., and Miranda, N.: Heterogeneous spatial and temporal pattern of surface elevation change and mass balance of the Patagonian ice fields between 2000 and 2016, The Cryosphere, 13, 2511–2535, doi:10.5194/tc-13-2511-2019, 2019.

Fan, Y., Ke, C.-Q., Zhou, X., Shen, X., Yu, X., and Lhakpa, D.: Glacier mass-balance estimates over High Mountain Asia from 2000 to 2021 based on ICESat-2 and NASADEM, J. Glaciol., 1–13, doi:10.1017/jog.2022.78, 2022.

Harrison, W. D. and Post, A. S.: How much do we really know about glacier surging?, Ann. Glaciol., 36, 1–6, doi:10.3189/172756403781816185, 2003.

Kochtiziky, W., Jiskoot, H., Copland, L., ENDERLIN, E., McNabb, R., KREUTZ, K., and MAIN, B.: Terminus advance, kinematics and mass redistribution during eight surges of Donjek Glacier, St. Elias Range, Canada, 1935 to 2016, J. Glaciol., 65, 565–579, doi:10.1017/jog.2019.34, 2019.

Raymond, C. F.: How do glaciers surge? A review, J. Geophys. Res., 92, 9121, doi:10.1029/JB092iB09p09121, 1987.

Rott, H., Sturm, K., and Miller, H.: Active and passive microwave signatures of Antarctic firn by means of field measurements and satellite data, Ann. Glaciol., 17, 337–343, doi:10.3189/S0260305500013070, 1993.

Truffer, M., Kääb, A., Harrison, W. D., Osipova, G. B., Nosenko, G. A., Espizua, L., Gilbert, A., Fischer, L., Huggel, C., Craw Burns, P. A., and Lai, A. W.: Glacier surges, in: Snow and Ice-Related Hazards, Risks, and Disasters, Elsevier, 417–466, doi:10.1016/B978-0-12-817129-5.00003-2, 2021.